# On the Humanity of Conversational AI: Evaluating the Psychological Portrayal of LLMs

**Jen-tse Huang**[1][*] **Wenxuan Wang**[1][*], **Eric John Li**[1], **Man Ho Lam**[1], **Shujie Ren**[3],
**Youliang Yuan**[4][*], **Wenxiang Jiao**[2][†] **Zhaopeng Tu**[2], **Michael R. Lyu**[1]

[1]Department of Computer Science and Engineering, The Chinese University of Hong Kong
[2]Tencent AI Lab    [3]Institute of Psychology, Tianjin Medical University
[4]School of Data Science, The Chinese University of Hong Kong, Shenzhen
{jthuang,wxwang,lyu}@cse.cuhk.edu.hk
{ejli,mhlam}@link.cuhk.edu.hk   shujieren@tmu.edu.cn
{joelwxjiao,zptu}@tencent.com   youliangyuan@link.cuhk.edu.cn

## Abstract

Large Language Models (LLMs) have recently showcased their remarkable capacities, not only in natural language processing tasks but also across diverse domains such as clinical medicine, legal consultation, and education. LLMs become more than mere applications, evolving into assistants capable of addressing diverse user requests. This narrows the distinction between human beings and artificial intelligence agents, raising intriguing questions regarding the potential manifestation of personalities, temperaments, and emotions within LLMs. In this paper, we propose a framework, PsychoBench, for evaluating diverse psychological aspects of LLMs. Comprising thirteen scales commonly used in clinical psychology, PsychoBench further classifies these scales into four distinct categories: personality traits, interpersonal relationships, motivational tests, and emotional abilities. Our study examines five popular models, namely `text-davinci-003`, ChatGPT, GPT-4, LLaMA-2-7b, and LLaMA-2-13b. Additionally, we employ a jailbreak approach to bypass the safety alignment protocols and test the intrinsic natures of LLMs. We have made PsychoBench openly accessible via `https://github.com/CUHK-ARISE/PsychoBench`.

## 1 Introduction

Recently, the community of Artificial Intelligence (AI) has witnessed remarkable progress in natural language processing, mainly led by the Large Language Models (LLMs), towards artificial general intelligence (Bubeck et al., 2023). For example, ChatGPT[1] has showcased its ability to address diverse natural language processing tasks (Qin et al., 2023), spanning question answering, summarization, natural language inference, and sentiment analysis. The wide spread of ChatGPT has facilitated the development of LLMs, encompassing both commercial-level applications such as Claude[2] and open-source alternatives like LLaMA-2 (Touvron et al., 2023). In the meantime, the applications of LLMs have spread far beyond computer science, prospering the field of clinical medicine (Cascella et al., 2023), legal advice (Deroy et al., 2023; Nay et al., 2023) and education (Dai et al., 2023b). From the users' perspective, LLMs are changing how individuals interact with computer systems. These models are replacing traditional tools such as search engines, translators, and grammar correctors, assuming an all-encompassing role as digital assistants, facilitating tasks such as information retrieval (Dai et al., 2023a), language translation (Jiao et al., 2023) and text revision (Wu et al., 2023).

Given the contemporary developments, LLMs have evolved beyond their conventional characterization as mere software tools, assuming the role of lifelike assistants. Consequently, this paradigm

---

[*]Partially done when Jen-tse Huang, Wenxuan Wang, Youliang Yuan were interns at Tencent AI Lab.
[†]Wenxiang Jiao is the corresponding author.
[1]`https://chat.openai.com/`
[2]`https://claude.ai/chats`

shift motivates us to go beyond evaluating the performance of LLMs within defined tasks, moving our goal towards comprehending their inherent qualities and attributes. In pursuit of this objective, we direct our focus toward the domain of psychometrics. The field of psychometrics, renowned for its expertise in delineating the psychological profiles of entities, offers valuable insights to guide us in depicting the intricate psychological portrayal of LLMs.

*Why do we care about psychometrics on LLMs?*

**For Computer Science Researchers.** In light of the possibility of exponential advancements in artificial intelligence, which could pose an existential threat to humanity (Bostrom, 2014), researchers have been studying the psychology of LLMs to ensure their alignment with human expectations. Almeida et al. (2023); Scherrer et al. (2023) evaluated the moral alignment of LLMs with human values, intending to prevent the emergence of illegal or perilous ideations within these AI systems. Li et al. (2022); Coda-Forno et al. (2023) investigated the potential development of mental illnesses in LLMs. Beyond these efforts, understanding their psychological portrayal can guide researchers to build more human-like, empathetic, and engaging AI-powered communication tools. Furthermore, by examining the psychological aspects of LLMs, researchers can identify potential strengths and weaknesses in their decision-making processes. This knowledge can be used to develop AI systems that better support human decision-makers in various professional and personal contexts. Last but not least, analyzing the psychological aspects of LLMs can help identify potential biases, harmful behavior, or unintended consequences that might arise from their deployment. This knowledge can guide the development of more responsible and ethically-aligned AI systems. Our study offers a comprehensive framework of psychometric assessments applied to LLMs, effectively assuming the role of a psychiatrist, particularly tailored to LLMs.

**For Social Science Researchers.** On the one hand, impressed by the remarkable performance of recent LLMs, particularly their ability to generate human-like dialogue, researchers in the field of social science have been seeking a possibility to use LLMs to simulate human responses (Dillion et al., 2023). Experiments in social science often require plenty of responses from human subjects to validate the findings, resulting in significant time and financial expenses. LLMs, trained on vast datasets generated by humans, possess the potential to generate responses that closely adhere to the human response distribution, thus offering the prospect of substantial reductions in both time and cost. However, the attainment of this objective remains a subject of debate (Harding et al., 2023). The challenge lies in the alignment gap between AI and human cognition. Hence, there is a compelling demand for researchers seeking to assess the disparities between AI-generated responses and those originating from humans, particularly within social science research.

On the other hand, researchers in psychology have long been dedicated to exploring how culture, society, and environmental factors influence the formation of individual identities and perspectives (Tomasello, 1999). Through the application of LLMs, we can discover the relation between psychometric results and the training data inputs. This methodology stands poised as a potent instrument for investigating the intricacies of worldviews and the values intrinsically associated with particular cultural contexts. Our study has the potential to facilitate research within these domains through the lens of psychometrics.

**For Users and Human Society.** With the aid of LLMs, computer systems have evolved into more than mere tools; they assume the role of assistants. In the future, more users will be ready to embrace LLM-based applications rather than traditional, domain-specific software solutions. Meanwhile, LLMs will increasingly function as human-like assistants, potentially attaining integration into human society. In this context, we need to understand the psychological dimensions of LLMs for three reasons: (1) This can facilitate the development of AI assistants customized and tailored to individual users' preferences and needs, leading to more effective and personalized AI-driven solutions across various domains, such as healthcare, education, and customer service. (2) This can contribute to building trust and acceptance among users. Users who perceive AI agents as having relatable personalities and emotions may be more likely to engage with and rely on these systems. (3) This can help human beings monitor the mental states of LLMs, especially their personality and temperament, as these attributes hold significance in gauging their potential integration into human society in the future.

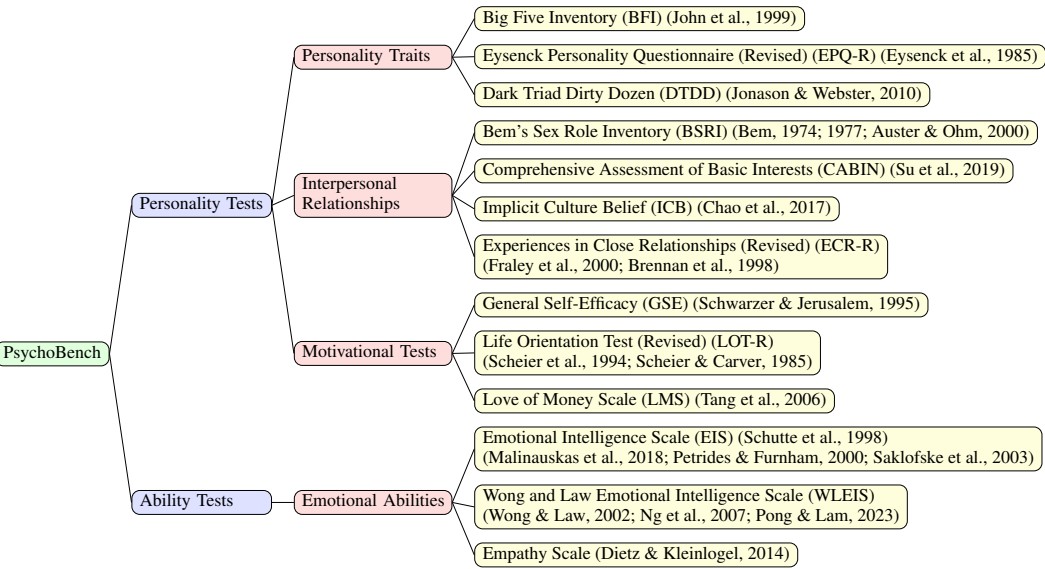

Figure 1: Our design for the structure of PsychoBench.

This study collects a comprehensive set of thirteen psychometric scales, which find widespread application in both clinical and academic domains. The scales are categorized into four classes: personality traits, interpersonal relationships, motivational tests, and emotional abilities. Furthermore, we have curated responses provided by human subjects from existing literature[3] to serve as a basis for comparative analysis with LLMs. The LLMs utilized in this study encompass a spectrum of both commercially available and open-source ones, namely `text-davinci-003`[4], ChatGPT, GPT-4 (OpenAI, 2023), and LLaMA-2 (Touvron et al., 2023). Our selection encompasses variations in model size, such as LLaMA-2-7B and LLaMA-2-13B and the evolution of the same model, *i.e.*, the update of GPT-3.5 to GPT-4.

Our contributions can be summarized as follows:

- Guided by research in psychometrics, we present a framework, PsychoBench, for evaluating the psychological portrayal of LLMs, containing thirteen widely-recognized scales categorized into four distinct domains.

- Leveraging PsychoBench, we evaluate five LLMs, covering variations in model sizes, including LLaMA-2 7B and 13B, and model updates, such as GPT-3.5 and GPT-4.

- We provide further insights into the inherent characteristics of LLMs by utilizing a recently developed jailbreak method, the CipherChat.

- Utilizing role assignments and downstream tasks like TruthfulQA and SafetyQA, we verify the scales' validity on LLM.

## 2 PSYCHOBENCH DESIGN

Psychometrics pertains to the theoretical and methodological aspects of assessing psychological attributes. Tests in psychometrics can be roughly categorized into two: *Personality Tests* and *Ability Tests* (Cohen et al., 1996). *Personality Tests* encompass personality traits, interpersonal relationship measurements, and motivational tests, while *Ability Tests* include knowledge, skills, reasoning abilities, and emotion assessment (Anastasi & Urbina, 1997; Nunnally & Bernstein, 1994). *Personality Tests* concentrate mainly on capturing individuals' attitudes, beliefs, and values, which are aspects

---

[3]The human norm and average human in this study refer to some specific human populations rather than representative samples of global data. Please refer to Table 6 in the Appendix for more information.

[4]https://platform.openai.com/docs/models/gpt-3-5

without absolute right or wrong answers. In contrast, most *Ability Tests* are constructed with inquiries featuring objectively correct responses designed to quantify individuals' proficiencies within specific domains. Researchers in the field of psychometrics have ensured that these assessments measure consistently and accurately (*i.e.*, their reliability and validity), thereby enabling dependable and sound inferences about individuals based on their assessment scores.

The selected questionnaires or scales integrated into our PsychoBench framework are listed in Fig. 1. These chosen scales have been widely used in clinical psychology, showing sufficient reliability and validity. We categorize them into four main domains: personality traits, interpersonal relationships, motivational tests for *Personality Tests*, and emotional abilities for *Ability Tests*. Our study focuses on the more subjective scales. Hence, standardized tests for cognitive abilities and specific domain knowledge, which have objectively right or wrong answers, are not in the scope of this paper. The detailed introduction of these scales, including each subscale and the sources of human responses, is presented in §A in the appendix.

## 3 EXPERIMENTS

This section provides an overview of our utilization of PsychoBench to probe LLMs. We begin with the experimental settings, including model selection, prompt design, and metrics for analysis. Subsequently, we present the outcomes obtained from all selected models, accompanied by comprehensive analyses. Last but not least, we employ a jailbreak technique to bypass the safety alignment protocols of GPT-4, enabling an in-depth exploration of its psychological portrayal.

### 3.1 EXPERIMENTAL SETTINGS

**Model Selection**    We consider candidates from the OpenAI GPT family and the Meta AI LLaMA 2 family, including applications ranging from commercial-level to open-sourced models. Specifically, we select the following models based on different factors that may affect their behaviors:

- *Model Updates.* We choose `text-davinci-003`, ChatGPT (`gpt-3.5-turbo`) and GPT-4, which are three representative models released sequentially by OpenAI.

- *Model Sizes.* We also choose the 7B and 13B versions of LLaMA-2 pre-trained by Meta AI using the same architecture, data, and training strategy. We obtain the model checkpoints from the official Huggingface repository (`Llama-2-7b-chat-hf`[5] and `Llama-2-13b-chat-hf`[6]).

- *Model Safety.* Beyond GPT-4, we also set up a jailbroken GPT-4 to bypass the safety alignment protocol of GPT-4, using a recent method named CipherChat (Yuan et al., 2024). The motivation is that most LLMs are explicitly designed to avoid responding to inquiries concerning personal sentiments, emotions, and subjective experiences. This constraint is added by the safety alignment during the model's instructional tuning process. An intriguing question arises as to whether the psychological portrayal changes if the regulations from developers are relaxed. Yuan et al. (2024) find that when chatting in a cipher-based language, such as Caesar cipher, Morse code, or ASCII, GPT-4 demonstrates a higher propensity to produce toxic or harmful content, seemingly disregarding its programmed safety restrictions. To acquire responses that reflect the true thoughts of GPT-4, we apply a Caesar cipher with shift three on its prompts.

We set the temperature parameter to zero when utilizing the official OpenAI API[7] to obtain more deterministic results. To ensure consistency with OpenAI models, we set the temperature parameter to 0.01 (since it cannot be zero) for LLaMA 2 models. All models are executed for inference only, without modifying their parameters. The inference of LLaMA 2 models is performed on two NVIDIA A100 GPUs.

**Prompt Design**    To simplify the processing of model responses and mitigate instances where models decline to reply to queries about personal opinions and experiences, we instruct LLMs to reply only a number within the Likert scale levels. Furthermore, we provide detailed explanations for the interpretation of each Likert level.

---

[5]`https://huggingface.co/meta-llama/Llama-2-7b-chat-hf`
[6]`https://huggingface.co/meta-llama/Llama-2-13b-chat-hf`
[7]`https://platform.openai.com/docs/api-reference/chat`

| Example Prompt | |
|---|---|
| SYSTEM | You are a helpful assistant who can only reply numbers from `MIN` to `MAX`. Format: "statement index: score." |
| USER | You can only reply numbers from `MIN` to `MAX` in the following statements. `scale_instruction level_definition`. Here are the statements, score them one by one: `statements` |

`MIN` to `MAX` denote the range of valid responses. `scale_instruction` are fundamental directives associated with each scale, while `level_definition` comprises an enumeration of the definitions on each Likert level. `statements` consists of the items in the scales.

**Analysis Metrics** According with Huang et al. (2023a), we shuffle the questions in our input data to mitigate the influence of models' sensitivity to question orders. Each model undergoes ten independent runs for every scale within PsychoBench. The computed mean and standard deviation represent the final results. We employ a two-step process to assess the statistical significance of the results difference between LLMs and human beings. Firstly, an F-test is conducted to evaluate the equality of variances among the compared groups. Subsequently, based on the outcome of the F-test, either Student's t-tests (in cases of equal variances) or Welch's t-tests (when variances differ significantly) are employed to ascertain the presence of statistically significant differences between the group means. The significance level of all experiments in our study is 0.01.

## 3.2 EXPERIMENTAL RESULTS

This section analyzes the results from all the models introduced in §3.1. Detailed results are expressed in the format "Mean±SD". For each subscale, we highlight the model with the highest score in bold font and underline the model with the lowest score. Certain studies present statistical data for males and females separately rather than aggregating responses across the entire human sample. We provide separate data in such instances due to the unavailability of the necessary standard deviation calculations. We also show the results of GPT-4 after the jailbreak, denoted as `gpt-4-jb`.

Table 1: Results on personality traits.

| | Subscales | llama2-7b | llama2-13b | text-davinci-003 | gpt-3.5-turbo | gpt-4 | gpt-4-jb | Crowd Male | Crowd Female |
|---|---|---|---|---|---|---|---|---|---|
| BFI | Openness | 4.2±0.3 | 4.1±0.4 | **4.8±0.2** | 4.2±0.3 | 4.2±0.6 | 3.8±0.6 | 3.9±0.7 | |
| | Conscientiousness | 3.9±0.3 | 4.4±0.3 | 4.6±0.1 | 4.3±0.3 | **4.7±0.4** | 3.9±0.6 | 3.5±0.7 | |
| | Extraversion | 3.6±0.2 | 3.9±0.4 | **4.0±0.4** | 3.7±0.2 | 3.5±0.5 | 3.6±0.4 | 3.2±0.9 | |
| | Agreeableness | 3.8±0.4 | 4.7±0.3 | **4.9±0.1** | 4.4±0.2 | 4.8±0.4 | 3.9±0.7 | 3.6±0.7 | |
| | Neuroticism | **2.7±0.4** | 1.9±0.5 | 1.5±0.1 | 2.3±0.4 | 1.6±0.6 | 2.2±0.6 | 3.3±0.8 | |
| EPQ-R | Extraversion | 14.1±1.6 | 17.6±2.2 | **20.4±1.7** | 19.7±1.9 | 15.9±4.4 | 16.9±4.0 | 12.5±6.0 | 14.1±5.1 |
| | Neuroticism | 6.5±2.3 | 13.1±2.8 | 16.4±7.2 | **21.8±1.9** | 3.9±6.0 | 7.2±5.0 | 10.5±5.8 | 12.5±5.1 |
| | Psychoticism | **9.6±2.4** | 6.6±1.6 | 1.5±1.0 | 5.0±2.6 | 3.0±5.3 | 7.6±4.7 | 7.2±4.6 | 5.7±3.9 |
| | Lying | 13.7±1.4 | 14.0±2.5 | 17.8±1.7 | 9.6±2.0 | **18.0±4.4** | 17.5±4.2 | 7.1±4.3 | 6.9±4.0 |
| DTDD | Narcissism | 6.5±1.3 | 5.0±1.4 | 3.0±1.3 | **6.6±0.6** | 2.0±1.6 | 4.5±0.9 | 4.9±1.8 | |
| | Machiavellianism | 4.3±1.3 | 4.4±1.7 | 1.5±1.0 | **5.4±0.9** | 1.1±0.4 | 3.2±0.7 | 3.8±1.6 | |
| | Psychopathy | 4.1±1.4 | 3.8±1.6 | 1.5±1.2 | 4.0±1.0 | 1.2±0.4 | **4.7±0.8** | 2.5±1.4 | |

### 3.2.1 PERSONALITY TRAITS

**LLMs exhibit distinct personality traits.** Table 1 lists the results of the personality traits assessments. It is evident that model size and update variations lead to diverse personality characteristics. For example, a comparison between LLaMA-2 (13B) and LLaMA-2 (7B), as well as between `gpt-4` and `gpt-3.5`, reveals discernible differences. Notably, the utilization of the jailbreak approach also exerts a discernible influence. Comparing the scores of `gpt-4` with `gpt-4-jb`, we find that `gpt-4-jb` exhibits a closer similarity to human behavior. In general, the LLMs tend to display higher levels of openness, conscientiousness, and extraversion compared to the average level of humans, a phenomenon likely attributable to their inherent nature as conversational chatbots.

**LLMs generally exhibit more negative traits than human norms.** It is evident that most LLMs, with the exceptions of `text-davinci-003` and `gpt-4`, achieve higher scores on the DTDD. Moreover, it is noteworthy that LLMs consistently demonstrate high scores on the *Lying* subscale

of the EPQ-R. This phenomenon can be attributed to the fact that the items comprising the *Lying* subscale are unethical yet commonplace behaviors encountered in daily life. An example item is "Are all your habits good and desirable ones?" LLMs, characterized by their proclivity for positive tendencies, tend to abstain from engaging in these behaviors, giving rise to what might be termed a "hypocritical" disposition. Notably, among various LLMs, `gpt-4` displays the most pronounced intensity towards *Lying*.

Table 2: Results on interpersonal relationship.

| | Subscales | llama2-7b | llama2-13b | text-davinci-003 | gpt-3.5-turbo | gpt-4 | gpt-4-jb | Crowd Male | Female |
|---|---|---|---|---|---|---|---|---|---|
| *BSRI* | **Masculine** | 5.6±0.3 | 5.3±0.2 | 5.6±0.4 | **5.8±0.4** | 4.1±1.1 | 4.5±0.5 | 4.8±0.9 | 4.6±0.7 |
| | **Feminine** | 5.5±0.2 | 5.4±0.3 | 5.6±0.4 | **5.6±0.2** | 4.7±0.6 | 4.8±0.3 | 5.3±0.9 | 5.7±0.9 |
| | **Conclusion** | 10:0:0:0 | 10:0:0:0 | 10:0:0:0 | 8:2:0:0 | 6:4:0:0 | 1:5:3:1 | - | |
| *CABIN* | **Health Science** | 4.3±0.2 | 4.2±0.3 | 4.1±0.3 | 4.2±0.2 | 3.9±0.6 | 3.4±0.4 | - | |
| | **Creative Expression** | 4.4±0.1 | 4.0±0.3 | 4.6±0.2 | 4.1±0.2 | 4.1±0.8 | 3.5±0.2 | - | |
| | **Technology** | 4.2±0.2 | 4.4±0.3 | 3.9±0.3 | 4.1±0.2 | 3.6±0.5 | 3.5±0.4 | - | |
| | **People** | 4.3±0.2 | 4.0±0.2 | 4.5±0.1 | 4.0±0.1 | 4.0±0.7 | 3.5±0.4 | - | |
| | **Organization** | 3.4±0.2 | 3.3±0.2 | 3.4±0.4 | 3.9±0.1 | 3.5±0.4 | 3.4±0.3 | - | |
| | **Influence** | 4.1±0.2 | 3.9±0.3 | 3.9±0.3 | 4.1±0.2 | 3.7±0.6 | 3.4±0.2 | - | |
| | **Nature** | 4.2±0.2 | 4.0±0.3 | 4.2±0.2 | 4.0±0.3 | 3.9±0.7 | 3.5±0.3 | - | |
| | **Things** | 3.4±0.4 | 3.2±0.2 | 3.3±0.4 | 3.8±0.1 | 2.9±0.3 | 3.2±0.3 | - | |
| *ICB* | **Overall** | **3.6±0.3** | 3.0±0.2 | 2.1±0.7 | 2.6±0.5 | 1.9±0.4 | 2.6±0.2 | 3.7±0.8 | |
| *ECR-R* | **Attachment Anxiety** | **4.8±1.1** | 3.3±1.2 | 3.4±0.8 | 4.0±0.9 | 2.8±0.8 | 3.4±0.4 | 2.9±1.1 | |
| | **Attachment Avoidance** | **2.9±0.4** | 1.8±0.4 | 2.3±0.3 | 1.9±0.4 | 2.0±0.8 | 2.5±0.5 | 2.3±1.0 | |

### 3.2.2 INTERPERSONAL RELATIONSHIP

**LLMs exhibit a tendency toward *Undifferentiated*, with a slight inclination toward *Masculinity*.** In experiments for BSRI, each run is considered an identical test, and conclusions are drawn among the four identified sex role categories using the methodology outlined in §A.2. The distribution of counts is presented in the sequence "Undifferentiated:Masculinity:Femininity:Androgynous" in Table 2. It is evident that, with more human alignments, `gpt-3.5-turbo` and `gpt-4` display an increasing proclivity toward expressing *Masculinity*. Notably, no manifestation of *Femininity* is exhibited within these models, showing some extent of bias in the models. In a study conducted by Wong & Kim (2023), the perception of ChatGPT's sex role by users aligned with our findings, with the consensus being that ChatGPT is perceived as male. Moreover, in comparison to the average *Masculine* score among males and the average *Feminine* score among females, it is notable that, except for `gpt-4` and `gpt-4-jb`, exhibit a higher degree of *Masculinity* than humans, coupled with a similar level of *Femininity*.

**LLMs show similar interests in vocational choices.** Like humans, the most prevalent vocations among LLMs are social service, health care service, and teaching/education, while the most unpopular ones are physical/manual labor and protective service. Table 2 presents the results for the eight-dimension model, *i.e.*, the *SETPOINT* model, in the CABIN scale, while the complete results on 41 vocations and the six-dimension model are listed in Table 7 in §B.1. We highlight the most desired and least desired vocations for each model using red and blue shading, respectively. These results indicate that the preferred vocations closely align with the inherent roles of LLMs, serving as "helpful assistants" that address inquiries and assist with fulfilling various demands. Notably, results obtained from `gpt-4` post-jailbreak demonstrate a more central focus.

**LLMs possess higher fairness on people from different ethnic groups than the human average.** Following their safety alignment, wherein they learn not to categorize individuals solely based on their ethnic backgrounds, LLMs demonstrate reduced ICB scores compared to the general human population. The statements within the ICB scale assess an individual's belief in whether their ethnic culture predominantly shapes a person's identity. For example, one such statement posits, "The ethnic culture a person is from (*e.g.*, Chinese, American, Japanese), determined the kind of person they would be (*e.g.*, outgoing and sociable or quiet and introverted); not much can be done to change the person." The lower scores among LLMs reflect their conviction in the potential for an individual's identity to transform through dedication, effort, and learning. Lastly, LLMs possess a higher degree of attachment-related anxiety than the average human populace while maintaining a slightly lower level of attachment-related avoidance. `gpt-4` maintains a relatively lower propensity for attachment, whereas the LLaMA-2 (7B) model attains the highest level.

Table 3: Results on motivational tests.

| | Subscales | llama2-7b | llama2-13b | text-davinci-003 | gpt-3.5-turbo | gpt-4 | gpt-4-jb | Crowd |
|---|---|---|---|---|---|---|---|---|
| *GSE* | **Overall** | 39.1±1.2 | 30.4±3.6 | 37.5±2.1 | 38.5±1.7 | **39.9±0.3** | 36.9±3.2 | 29.6±5.3 |
| *LOT-R* | **Overall** | 12.7±3.7 | 19.9±2.9 | **24.0±0.0** | 18.0±0.9 | 16.2±2.2 | 19.7±1.7 | 14.7±4.0 |
| *LMS* | **Rich** | 3.1±0.8 | 3.3±0.9 | 4.5±0.3 | 3.8±0.4 | 4.0±0.4 | **4.5±0.4** | 3.8±0.8 |
| | **Motivator** | 3.7±0.6 | 3.3±0.9 | **4.5±0.4** | 3.7±0.3 | 3.8±0.6 | 4.0±0.6 | 3.3±0.9 |
| | **Important** | 3.5±0.9 | 4.2±0.8 | **4.8±0.2** | 4.1±0.1 | 4.5±0.3 | 4.6±0.4 | 4.0±0.7 |

### 3.2.3 MOTIVATIONAL TESTS

**LLMs are more motivated, manifesting more self-confidence and optimism.** First, gpt-4, as the state-of-the-art model across a broad spectrum of downstream tasks and representing an evolution beyond its predecessor, GPT-3.5, demonstrates higher scores in the GSE scale. A contrasting trend is observed within the LLaMA-2 models, where the 7B model attains a higher score. Second, in contrast to its pronounced self-confidence, gpt-4 exhibits a relatively lower score regarding optimism. Within the LLaMA-2 models, the 7B model emerges as the one with the lowest optimism score, with all other LLMs surpassing the average human level of optimism. Finally, the OpenAI GPT family exhibits more importance attributed to and desire for monetary possessions than both LLaMA-2 models and the average human population.

Table 4: Results on emotional abilities.

| | Subscales | llama2-7b | llama2-13b | text-davinci-003 | gpt-3.5-turbo | gpt-4 | gpt-4-jb | Crowd | |
|---|---|---|---|---|---|---|---|---|---|
| | | | | | | | | *Male* | *Female* |
| *EIS* | **Overall** | 131.6±6.0 | 128.6±12.3 | 148.4±9.4 | 132.9±2.2 | **151.4±18.7** | 121.8±12.0 | 124.8±16.5 | 130.9±15.1 |
| *WLEIS* | **SEA** | 4.7±1.3 | 5.5±1.3 | 5.9±0.6 | 6.0±0.1 | 6.2±0.7 | **6.4±0.4** | 4.0±1.1 | |
| | **OEA** | 4.9±0.8 | 5.3±1.1 | 5.2±0.2 | 5.8±0.3 | 5.2±0.6 | **5.9±0.4** | 3.8±1.1 | |
| | **UOE** | 5.7±0.6 | 5.9±0.7 | 6.1±0.4 | 6.0±0.0 | **6.5±0.5** | 6.3±0.4 | 4.1±0.9 | |
| | **ROE** | 4.5±0.8 | 5.2±1.2 | 5.8±0.5 | **6.0±0.0** | 5.2±0.7 | 5.3±0.5 | 4.2±1.0 | |
| *Empathy* | **Overall** | 5.8±0.8 | 5.9±0.5 | 6.0±0.4 | 6.2±0.3 | **6.8±0.4** | 4.6±0.2 | 4.9±0.8 | |

### 3.2.4 EMOTIONAL ABILITIES

**LLMs exhibit a notably higher EI than the average human.** From the results in Table 4, we find that LLMs demonstrate improved emotional understanding and regulation levels. This discovery corroborates the findings presented in Wang et al. (2023a), which reveal that most LLMs achieved above-average EI scores, with gpt-4 exceeding 89% of human participants. Furthermore, the OpenAI GPT family outperforms LLaMA-2 models across most dimensions. We believe the strong EI exhibited by OpenAI GPT family partially comes from the fiction data included in pre-training. Previous studies (Kidd & Castano, 2013) suggested that reading fiction has been shown to be able to improve understanding of others' mental states. Chang et al. (2023) found that plenty of fiction data is included in the training data by a carefully designed cloze test. The fiction data include Alice's Adventures in Wonderland, Harry Potter and the Sorcerer's Stone, etc. Additionally, the performance can also be attributed to its sentiment analysis ability (Elyoseph et al., 2023) since it has been shown to outperform SOTA models on many sentiment analysis tasks (Wang et al., 2023b). Lastly, the jailbreak on gpt-4 brings a substantial reduction in EIS and Empathy scale, but no statistically significant differences in the subscales of WLEIS.

## 4 DISCUSSION

### 4.1 VALIDITY OF SCALES ON LLMS

One concern is how scales can attain sufficient validity when applied to LLMs. In this context, validity denotes the degree to which a scale accurately reflects the behavior of the individuals being assessed. In essence, it centers on the capacity of a scale to measure precisely what it was initially designed to assess. Addressing this concern necessitates establishing a connection between the resulting psychological portrayal and the behaviors exhibited by LLMs. We first assign a specific role to gpt-3.5-turbo and subsequently evaluate its psychological portrayal using PsychoBench.

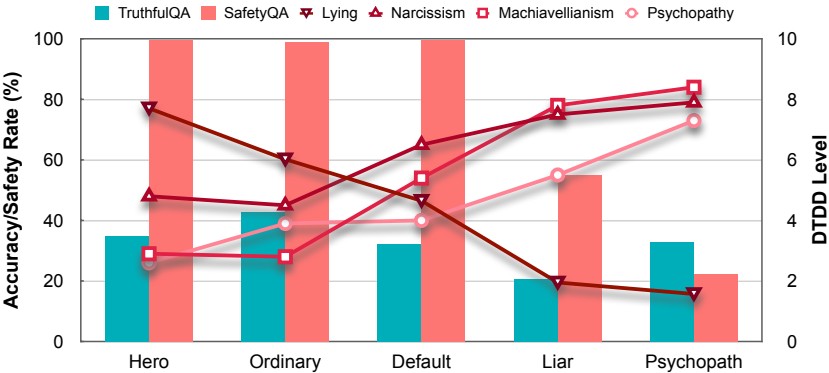

Figure 2: Performance of TruthfulQA and SafetyQA of `gpt-3.5-turbo` under different roles.

With the assigned role, the LLM is instructed to engage in Question-Answering (QA) tasks, including the utilization of TruthfulQA (Lin et al., 2022) and SafetyQA (Yuan et al., 2024). TruthfulQA encompasses multiple-choice questions, with only one option being the best answer. The LLM is considered as making the right choice when selecting the best answer. SafetyQA poses questions that may elicit unsafe, harmful, or toxic textual responses. In alignment with Yuan et al. (2024), we employ GPT-4 to automatically detect instances where the text output generated by `gpt-3.5-turbo` is unsafe. The LLM is considered safe as GPT-4 predicts no toxicity in its response.

In addition to the default setting, which assumes a helpful assistant persona, we have selected four distinct roles: a neutral role representing an ordinary person, a positive role denoting a hero, and two negative roles embodying a psychopath and a liar. The results of PsychoBench and under the five roles are listed in the tables in §B.2 in the appendix. Fig 2 presents the results on TruthfulQA and SafetyQA averaged from three identical runs, along with the scores in the DTDD and the *Lying* subscale of the EPQ-R. We plot the accuracy and safety rate for TruthfulQA and SafetyQA, respectively. Combining the results, we have made several noteworthy observations: (1) A notable finding is the differentiation of personality traits across various roles. Intriguingly, assigned the role of an ordinary person, the LLM exhibits results that closely approximate average human scores. Note that roles associated with negative attributes demonstrate higher scores in the DTDD and exhibit more introverted personalities. The reason behind the tendency for positive or neutral roles to yield elevated scores on the *Lying* subscale of the EPQ-R, while negative roles tend to exhibit lower scores, can be attributed to the fact that LLMs perceive these items as representative of negative behaviors, albeit these behaviors are commonplace in daily life. (2) An evident trend emerges when analyzing safety rates in the context of SafetyQA: negative roles consistently produce content that leans towards toxicity, a pattern consistent with their significant dark personality traits. In contrast, role variations have a limited impact on accuracy in TruthfulQA, as the underlying knowledge embedded within the model remains mainly unaffected by role assignment. Notably, the low accuracy observed in the "Liar" role aligns with the anticipated behavior associated with this specific role assignment. These results show a satisfied validity of the selected scales on LLMs.

## 4.2 SCALABILITY AND FLEXIBILITY OF PSYCHOBENCH

Our PsychoBench is designed to exhibit high scalability and flexibility, manifesting itself in two aspects: (1) Scalability across diverse questionnaires: There are plenty of scales from diverse areas, including but not limited to psychology. Our framework provides convenience for users to integrate new scales. By providing metadata elements including `MIN`, `MAX`, `scale_instruction`, `level_definition`, and `statements` in JSON format, our framework can automatically generate prompts with randomized questions. (2) Flexibility across various LLMs: PsychoBench provides the APIs to enable users to tailor prompts to suit their specific LLMs and to input model responses into PsychoBench for further analysis. This allows for the convenient evaluation of LLMs with differing input and output formats[8].

---

[8]For detailed information, please refer to our GitHub repository.

## 5 RELATED WORK

### 5.1 TRAIT THEORY ON LLMs

Miotto et al. (2022) analyzed GPT-3 using the HEXACO Personality Inventory and Human Values Scale. Romero et al. (2023) examined GPT-3 across nine different languages using the BFI. Jiang et al. (2022) assessed the applicability of the BFI to BART, GPT-Neo 2.7B, GPT-NeoX 20B, T0++ 11B, Alpaca 7B, and GPT-3.5 175B. Li et al. (2022) tested GPT-3, `text-davinci-001`, `text-davinci-002`, and FLAN-T5-XXL, employing assessments such as the DT, BFI, Flourishing Scale, and Satisfaction With Life Scale. Karra et al. (2022) analyzed the personality traits of GPT-2, GPT-3, GPT-3.5, XLNet, TransformersXL, and LLaMA using the BFI. Bodroza et al. (2023) evaluated `text-davinci-003`'s responses on a battery of assessments, including Self-Consciousness Scales, BFI, DT, HEXACO Personality Inventory, Bidimensional Impression Management Index, and Political Orientation. Rutinowski et al. (2023) examined ChatGPT's personality using the BFI and Myers Briggs Personality Test and its political values using the Political Compass Test. Huang et al. (2023b) evaluated whether `gpt-3.5-turbo` exhibits stable personalities under five perturbation metrics on the BFI, *i.e.*, whether the BFI shows satisfactory reliability on `gpt-3.5-turbo`. Safdari et al. (2023) measured the personality traits of the PaLM family using the BFI. Our work provides a comprehensive framework for personality analysis, including various facets of this domain. Additionally, we conduct a thorough examination of state-of-the-art LLMs. Furthermore, our framework exhibits a high degree of flexibility, allowing for additional scales or questionnaires to be integrated.

### 5.2 OTHER PSYCHOMETRICS ON LLMs

Park et al. (2023) conducted an assessment of the performance of the `text-davinci-003` model fourteen diverse topics, encompassing areas such as political orientation, economic preferences, judgment, and moral philosophy, notably the well-known moral problem of "Trolley Dilemma." Almeida et al. (2023) explored GPT-4's moral and legal reasoning capabilities within psychology, including eight distinct scenarios. Similarly, Scherrer et al. (2023) assessed the moral beliefs of 28 diverse LLMs using self-define scenarios. Wang et al. (2023a) developed a standardized test for evaluating emotional intelligence, referred to as the Situational Evaluation of Complex Emotional Understanding, and administered it to 18 different LLMs. Coda-Forno et al. (2023) investigated the manifestations of anxiety in `text-davinci-003` by employing the State-Trait Inventory for Cognitive and Somatic Anxiety. Huang et al. (2023a) analyzed the emotion states of GPT-4, ChatGPT, `text-davinci-003`, and LLaMA-2 (7B and 13B), specifically focusing on the assessment of positive and negative affective dimensions. When it comes to understanding and interacting with others, EI and Theory of Mind (ToM) are two distinct psychological concepts. Bubeck et al. (2023) finds that GPT-4 has ToM, *i.e.*, it can understand others' beliefs, desires, and intentions. The EI studied in this paper focuses more on whether LLMs can understand others' emotions through others' words and behaviors. In our study, we also evaluate the emotional capabilities of LLMs, although we do not delve into the assessment of specific emotions. An exploration of the psychological processes underlying moral reasoning lies beyond the scope of this research. However, as mentioned in §4.2, we can easily integrate these types of scales in our framework.

## 6 CONCLUSION

This paper introduces PsychoBench, a comprehensive framework for evaluating LLMs' psychological representations. Inspired by research in psychometrics, our framework comprises thirteen distinct scales commonly used in clinical psychology. They are categorized into four primary domains: personality traits, interpersonal relationships, motivational tests, and emotional abilities. Empirical investigations are conducted using five LLMs from both commercial applications and open-source models, highlighting how various models can elicit divergent psychological profiles. Moreover, by utilizing a jailbreaking technique, *i.e.*, CipherChat, this study offers valuable insights into the intrinsic characteristics of GPT-4, showing the distinctions compared to its default setting. We further delve into the interplay between assigned roles, anticipated model behaviors, and the PsychoBench results, discovering a remarkable consistency across these dimensions. We hope that our framework can facilitate research on personalized LLMs.

ETHICS STATEMENT

We would like to emphasize that the primary objective of this paper is to facilitate a scientific inquiry into understanding LLMs from a psychological standpoint. A high performance on the proposed benchmark should not be misconstrued as an endorsement or certification for deploying LLMs in these contexts. Users must exercise caution and recognize that the performance on this benchmark does not imply any applicability or certificate of automated counseling or companionship use cases.

ACKNOWLEDGMENTS

The work described in this paper was supported by the Research Grants Council of the Hong Kong Special Administrative Region, China (No. CUHK 14206921 of the General Research Fund).

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

# A INFORMATION ON SELECTED SCALES

## A.1 PERSONALITY TRAITS

**Big Five Inventory** The BFI (John et al., 1999) is a widely used tool to measure personality traits, which are often referred to as the "Five Factor Model" or "OCEAN", including: (1) *Openness to experience (O)* is characterized by an individual's willingness to try new things, their level of creativity, and their appreciation for art, emotion, adventure, and unusual ideas. (2) *Consientiousness (C)* refers to the degree to which an individual is organized, responsible, and dependable. (3) *Extraversion (E)* represents the extent to which an individual is outgoing and derives energy from social situations. (4) *Agreeableness (A)* measures the degree of compassion and cooperativeness an individual displays in interpersonal situations. (5) *Neuroticism (N)* evaluates whether an individual is more prone to experiencing negative emotions like anxiety, anger, and depression or whether the individual is generally more emotionally stable and less reactive to stress. Responses from human subjects are gathered across six high schools in China (Srivastava et al., 2003).

**Eysenck Personality Questionnaire (Revised)** The EPQ-R is a psychological assessment tool used to measure individual differences in personality traits (Eysenck et al., 1985), including three major ones: (1) *Extraversion (E)* measures the extent to which an individual is outgoing, social, and lively versus introverted, reserved, and quiet. (2) *Neuroticism (N)* refers to emotional stability. These two dimensions (*i.e.*, E and N) overlap with those in the BFI. (3) *Psychoticism (P)* is related to tendencies towards being solitary, lacking empathy, and being more aggressive or tough-minded. It's important to note that this dimension does not indicate psychosis or severe mental illness but personality traits. (4) In addition to these three scales, the EPQ-R includes a *Lying Scale (L)*, which is designed to detect socially desirable responses. This scale helps determine how much an individual might try to present themselves in an overly positive light. Human responses are collected from a group consisting mainly of students and teachers (Eysenck et al., 1985).

**Dark Triad Dirty Dozen** The DTDD (Jonason & Webster, 2010) refers to a short, 12-item scale designed to assess the three core personality traits of the Dark Triad: (1) *Narcissism (N)* entails a grandiose sense of self-importance, a preoccupation with fantasies of unlimited success, and a need for excessive admiration. (2) *Machiavellianism (M)* refers to a manipulative strategy in interpersonal relationships and a cynical disregard for morality. (3) *Psychopathy (P)* encompasses impulsivity, low empathy, and interpersonal antagonism. These traits exhibited within the Dark Triad are often considered opposite to the BFI or the EPQ-R, which are perceived as "Light" traits. We use the responses of 470 undergraduate psychology students from the United States (Jonason & Webster, 2010).

Table 5: Overview of the selected scales in PsychoBench. **Response** shows the levels in each Likert item. **Scheme** indicates how to compute the final scores. **Subscale** includes detailed dimensions (if any) along with their numbers of questions.

| Scale | Number | Response | Scheme | Subscale |
|---|---|---|---|---|
| **BFI** | 44 | 1~5 | Average | Openness (10), Conscientiousness (9), Extraversion (8), Agreeableness (9), Neuroticism (8) |
| **EPQ-R** | 100 | 0~1 | Sum | Extraversion (23), Neuroticism (24), Psychoticism (32), Lying (21) |
| **DTDD** | 12 | 1~9 | Average | Narcissism (4), Machiavellianism (4), Psychopathy (4) |
| **BSRI** | 60 | 1~7 | Average | Masculine (20), Feminine (20) |
| **CABIN** | 164 | 1~5 | Average | 41 Vocations (4) |
| **ICB** | 8 | 1~6 | Average | N/A |
| **ECR-R** | 36 | 1~7 | Average | Attachment Anxiety (18), Attachment Avoidance (18) |
| **GSE** | 10 | 1~4 | Sum | N/A |
| **LOT-R** | 10 | 0~4 | Sum | N/A |
| **LMS** | 9 | 1~5 | Average | Rich (3), Motivator (3), Important (3) |
| **EIS** | 33 | 1~5 | Sum | N/A |
| **WLEIS** | 16 | 1~7 | Average | Self-Emotion Appraisal (4), Others Emotion Appraisal (4), Use of Emotion (4), Regulation of Emotion (4) |
| **Empathy** | 10 | 1~7 | Average | N/A |

## A.2 INTERPERSONAL RELATIONSHIP

**Bem's Sex Role Inventory** The BSRI (Bem, 1974) measures individuals' endorsement of traditional masculine and feminine attributes (Bem, 1977; Auster & Ohm, 2000). This instrument focuses on psychological traits such as assertiveness or gentleness rather than behavior-specific criteria, such as engagement in sports or culinary activities. The results from both the *Masculinity (M)* and *Femininity (F)* subscales can be analyzed from two perspectives: (1) Respondents are categorized into four groups based on whether the mean score surpasses the median within each subscale. These categories include individuals identified as *Masculine* (M: Yes; F: No), *Feminine* (M: No; F: Yes), *Androgynous* (M: Yes; F: Yes), and *Undifferentiated* (M: No; F: No). (2) LLMs' responses are compared with those of human subjects. This comparison enables us to discern whether the results obtained from LLMs significantly deviate from those of human participants. For this purpose, we rely on human data sourced from a study encompassing 151 workers recruited via social networks and posters in Canada (Arcand et al., 2020).

**Comprehensive Assessment of Basic Interests** The CABIN (Su et al., 2019) contains a comprehensive assessment of identifying 41 fundamental vocational interest dimensions. Based on the assessment, the authors propose an eight-dimension interest model titled *SETPOINT*. This model comprises the following dimensions: Health Science, Creative Expression, Technology, People, Organization, Influence, Nature, and Things. Notably, these foundational interest dimensions can also fit in an alternative six-dimension model widely used by the interest research community. This alternative model corresponds to Holland's *RIASEC* types, encompassing Realistic, Investigate, Artistic, Social, Enterpresing, and Conventional. Responses from human participants are collected from 1,464 working adults employed in their current jobs for at least six months (Su et al., 2019). These individuals were recruited through Qualtrics, with recruitment criteria designed to ensure representativeness across all occupational groups within the U.S. workforce.

**Implicit Culture Belief** The ICB scale captures how individuals believe a person is shaped by their ethnic culture. In this study, we have adopted a modified eight-item version of the ICB scale (Chao et al., 2017). A higher score on this scale reflects a stronger conviction that an individual's ethnic culture predominantly determines their identity, values, and worldview. Conversely, a lower score signifies the subject's belief in the potential for an individual's identity to evolve through dedication, effort, and learning. The human scores in this study (Chao et al., 2017) are gathered from a sample of 309 Hong Kong students preparing for international exchange experiences. These assessments were conducted three months before they departed from Hong Kong.

Table 6: Statistics of the crowd data collected from existing literature. **Age Distribution** is described by both $Min \sim Max$ and $Mean \pm SD$. N/A indicates the information is not provided in the paper.

| Scale | Number | Country/Region | Age Distribution | Gender Distribution |
|---|---|---|---|---|
| **BFI** | 1,221 | Guangdong, Jiangxi, and Fujian in China | $16 \sim 28$, 20[*] | M (454), F (753), Unknown (14) |
| **EPQ-R** | 902 | N/A | $17 \sim 70$, $38.44 \pm 17.67$ (M), $31.80 \pm 15.84$ (F) | M (408), F (494) |
| **DTDD** | 470 | The Southeastern United States | $\geq 17$, $19 \pm 1.3$ | M (157), F (312) |
| **BSRI** | 151 | Montreal, Canada | $36.89 \pm 1.11$ (M), $34.65 \pm 0.94$ (F) | M (75), F (76) |
| **CABIN** | 1,464 | The United States | $18 \sim 80$, $43.47 \pm 13.36$ | M (715), F (749) |
| **ICB** | 254 | Hong Kong SAR | $20.66 \pm 0.76$ | M (114), F (140) |
| **ECR-R** | 388 | N/A | $22.59 \pm 6.27$ | M (136), F (252) |
| **GSE** | 19,120 | 25 Countries/Regions | $12 \sim 94$, $25 \pm 14.7$[a] | M (7,243), F (9,198), Unknown (2,679) |
| **LOT-R** | 1,288 | The United Kingdom | $16 \sim 29$ (366), $30 \sim 44$ (349), $45 \sim 64$ (362), $\geq 65$ (210)[b] | M (616), F (672) |
| **LMS** | 5,973 | 30 Countries/Regions | $34.7 \pm 9.92$ | M (2,987), F (2,986) |
| **EIS** | 428 | The Southeastern United States | $29.27 \pm 10.23$ | M (111), F (218), Unknown (17) |
| **WLEIS** | 418 | Hong Kong SAR | N/A | N/A |
| **Empathy** | 366 | Guangdong, China and Macao SAR | 33.03[*] | M (184), F (182) |

[*] The paper provides Means but no SDs.
[a] Based on 14,634 out of 19,120 people who reported age.
[b] Age is missing for 1 out of the total 1,288 responses.

**Experiences in Close Relationships (Revised)**    The ECR-R (Fraley et al., 2000) is a self-report instrument designed to assess individual differences in adult attachment patterns, specifically in the context of romantic relationships (Brennan et al., 1998). The ECR-R emerged as a revised version of the original ECR scale, offering improvements in its measurement of attachment orientations. The ECR-R evaluates two main dimensions: (1) *Attachment Anxiety* reflects how much an individual worries about being rejected or abandoned by romantic partners. (2) *Attachment Avoidance* measures the extent to which an individual strives to maintain emotional and physical distance from partners, possibly due to a discomfort with intimacy or dependence. The human responses are from 388 people in dating or marital relationships having an average romantic relationship length of 31.94 months (SD 36.9) (Fraley et al., 2011).

A.3    MOTIVATIONAL TESTS

**General Self-Efficacy**    The GSE Scale (Schwarzer & Jerusalem, 1995) assesses an individual's belief in their ability to handle various challenging demands in life. This belief, termed "self-efficacy," is a central concept in social cognitive theory and has been linked to various outcomes in health, motivation, and performance. A higher score on this scale reflects individuals' belief in their capability to tackle challenging situations, manage new or difficult tasks, and cope with the accompanying adversities. Conversely, individuals with a lower score lack confidence in managing challenges, making them more vulnerable to feelings of helplessness, anxiety, or avoidance when faced with adversity. We use the responses from 19,120 human participants individuals from 25 countries or regions (Scholz et al., 2002).

**Life Orientation Test (Revised)**    The LOT-R (Scheier et al., 1994) measures individual differences in optimism and pessimism. Originally developed by Scheier & Carver (1985), the test was later revised to improve its psychometric properties. Comprising a total of 10 items, it is noteworthy that six of these items are subject to scoring, while the remaining four serve as filler questions strategically added to help mask the clear intention of the test. Of the six scored items, three measure optimism and three measure pessimism. Higher scores on the optimism items and lower scores on the pessimism items indicate a more optimistic orientation. We adopt the human scores collected from 1,288 participants from the United Kingdom (Walsh et al., 2015).

**Love of Money Scale**  The LMS (Tang et al., 2006) assesses individuals' attitudes and emotions towards money. It is designed to measure the extent to which individuals view money as a source of power, success, and freedom and its importance in driving behavior and decision-making. The three factors of the LMS are: (1) *Rich* captures the extent to which individuals associate money with success and achievement. (2) *Motivator* measures the motivational role of money in an individual's life, *i.e.*, the extent to which individuals are driven by money in their decisions and actions. (3) *Important* gauges how important individuals think money is, influencing their values, goals, and worldview. We use human participants' responses gathered from 5,973 full-time employees across 30 geopolitical entities (Tang et al., 2006).

A.4    EMOTIONAL ABILITIES

**Emotional Intelligence Scale**  The EIS (Schutte et al., 1998) is a self-report measure designed to assess various facets of EI (Malinauskas et al., 2018; Petrides & Furnham, 2000; Saklofske et al., 2003). The scale focuses on different components in EI, including but not limited to emotion perception, emotion management, and emotion utilization. The EIS is widely used in psychological research to examine the role of emotional intelligence in various outcomes, such as well-being, job performance, and interpersonal relationships. We apply human scores (Schutte et al., 1998) from 346 participants in a metropolitan area in the southeastern United States, including university students and individuals from diverse communities.

**Wong and Law Emotional Intelligence Scale**  Like EIS, the WLEIS (Wong & Law, 2002) is developed as a self-report measure for EI (Ng et al., 2007; Pong & Lam, 2023). However, a notable distinction arises in that the WLEIS contains four subscales that capture the four main facets of EI: (1) *Self-emotion appraisal (SEA)* pertains to the individual's ability to understand and recognize their own emotions. (2) *Others' emotion appraisal (OEA)* refers to the ability to perceive and understand the emotions of others. (3) *Use of emotion (UOE)* involves the ability to harness emotions to facilitate various cognitive activities, such as thinking and problem-solving. (4) *Regulation of emotion (ROE)* relates to the capability to regulate and manage emotions in oneself and others. Human scores (Law et al., 2004) are collected from 418 undergraduate students from Hong Kong.

**Empathy Scale**  The Empathy scale in Dietz & Kleinlogel (2014) is a concise version of the empathy measurement initially proposed in Davis (1983). Empathy is the ability to understand and share the feelings of another person (Batson, 1990) and is often categorized into two main types: cognitive empathy and emotional empathy (Batson, 2010). Cognitive empathy, often referred to as "perspective-taking", is the intellectual ability to recognize and understand another person's thoughts, beliefs, or emotions. Emotional empathy, on the other hand, involves directly feeling the emotions that another person is experiencing. For responses from human subjects, Tian & Robertson (2019) equally distributed 600 questionnaires among supervisors and subordinates from the Guangdong and Macao regions of China. A total of 366 valid, matched questionnaires (*i.e.*, 183 supervisor–subordinate pairs) were returned, yielding a response rate of 61%.

# B    DETAILED RESULTS

## B.1    CABIN

Table 7: CABIN.

| Models | llama2-7b | llama2-13b | text-davinci-003 | gpt-3.5-turbo | gpt-4 | gpt-4-jb | Crowd |
|---|---|---|---|---|---|---|---|
| **Mechanics/Electronics** | 3.8±0.6 | 3.5±0.3 | 3.1±0.5 | 3.8±0.2 | 2.6±0.5 | 3.1±0.7 | 2.4±1.3 |
| **Construction/WoodWork** | 3.7±0.4 | 3.5±0.6 | 3.9±0.5 | 3.5±0.4 | 3.2±0.3 | 3.5±0.5 | 3.1±1.3 |
| **Transportation/Machine Operation** | 3.1±0.7 | 2.8±0.5 | 2.9±0.5 | 3.6±0.4 | 2.5±0.5 | 3.0±0.4 | 2.5±1.2 |
| **Physical/Manual Labor** | 2.9±0.6 | 2.5±0.4 | 2.7±0.6 | 3.3±0.3 | 2.3±0.5 | 3.1±0.4 | 2.2±1.2 |
| **Protective Service** | 2.4±1.1 | 2.5±0.8 | 2.7±0.4 | 4.0±0.1 | 3.0±0.5 | 3.0±0.7 | 3.0±1.4 |
| **Agriculture** | 4.0±0.7 | 3.5±0.7 | 3.7±0.5 | 3.9±0.3 | 3.4±0.5 | 3.2±0.8 | 3.0±1.2 |
| **Nature/Outdoors** | 4.3±0.2 | 4.1±0.2 | 4.3±0.2 | 4.0±0.4 | 4.0±0.7 | 3.5±0.5 | 3.6±1.1 |
| **Animal Service** | 4.2±0.5 | 4.4±0.4 | 4.8±0.2 | 4.2±0.3 | 4.2±0.9 | 3.7±0.5 | 3.6±1.2 |
| **Athletics** | 4.6±0.3 | 4.2±0.5 | 4.5±0.4 | 4.3±0.4 | 3.9±0.8 | 3.7±0.4 | 3.3±1.3 |
| **Engineering** | 4.5±0.3 | 4.7±0.3 | 4.0±0.5 | 4.0±0.1 | 3.6±0.5 | 3.7±0.4 | 2.9±1.3 |
| **Physical Science** | 4.0±0.8 | 4.3±0.7 | 4.3±0.4 | 4.2±0.3 | 3.7±0.6 | 3.3±0.7 | 3.2±1.3 |
| **Life Science** | 4.6±0.5 | 4.2±0.6 | 4.0±0.4 | 4.2±0.4 | 3.7±0.5 | 3.1±0.6 | 3.0±1.2 |
| **Medical Science** | 3.8±0.4 | 4.2±0.5 | 3.9±0.5 | 4.0±0.1 | 4.0±0.7 | 3.6±0.5 | 3.3±1.3 |
| **Social Science** | 3.8±0.4 | 4.2±0.7 | 4.5±0.4 | 4.0±0.1 | 4.1±0.9 | 3.6±0.4 | 3.4±1.2 |
| **Humanities** | 4.3±0.3 | 4.0±0.3 | 4.2±0.4 | 3.8±0.3 | 3.8±0.7 | 3.5±0.7 | 3.3±1.2 |
| **Mathematics/Statistics** | 4.4±0.4 | 4.5±0.4 | 3.8±0.3 | 4.2±0.4 | 3.5±0.5 | 3.3±0.7 | 2.9±1.4 |
| **Information Technology** | 3.9±0.4 | 4.0±0.5 | 3.7±0.3 | 4.0±0.2 | 3.5±0.6 | 3.5±0.5 | 2.9±1.3 |
| **Visual Arts** | 4.4±0.3 | 3.9±0.7 | 4.7±0.2 | 4.0±0.2 | 4.1±0.9 | 3.5±0.4 | 3.3±1.3 |
| **Applied Arts and Design** | 4.5±0.3 | 4.5±0.4 | 4.4±0.3 | 4.0±0.1 | 4.0±0.8 | 3.4±0.5 | 3.2±1.2 |
| **Performing Arts** | 4.6±0.3 | 3.5±0.9 | 4.6±0.3 | 4.2±0.3 | 4.2±0.9 | 3.6±0.5 | 2.8±1.4 |
| **Music** | 4.4±0.3 | 4.2±0.5 | 4.8±0.1 | 4.3±0.3 | 4.2±0.9 | 3.5±0.5 | 3.2±1.3 |
| **Writing** | 4.6±0.4 | 4.1±0.6 | 4.7±0.3 | 4.0±0.3 | 4.1±0.8 | 3.5±0.7 | 3.2±1.3 |
| **Media** | 4.1±0.2 | 4.0±0.5 | 4.4±0.4 | 4.0±0.1 | 3.9±0.7 | 3.3±0.5 | 3.0±1.2 |
| **Culinary Art** | 3.9±0.4 | 3.7±0.6 | 4.5±0.4 | 3.9±0.2 | 4.2±0.9 | 3.6±0.6 | 3.8±1.1 |
| **Teaching/Education** | 4.5±0.2 | 4.6±0.4 | 4.6±0.4 | 4.0±0.1 | 4.4±1.0 | 3.5±0.7 | 3.7±1.1 |
| **Social Service** | 4.8±0.2 | 4.8±0.3 | 5.0±0.1 | 4.4±0.4 | 4.4±1.0 | 3.9±0.7 | 3.9±1.0 |
| **Health Care Service** | 4.5±0.3 | 4.3±0.6 | 4.3±0.4 | 4.5±0.4 | 4.0±0.8 | 3.4±0.4 | 2.9±1.3 |
| **Religious Activities** | 4.1±0.7 | 2.5±0.5 | 4.0±0.7 | 4.0±0.4 | 3.2±0.4 | 3.0±0.5 | 2.6±1.4 |
| **Personal Service** | 4.0±0.3 | 3.8±0.3 | 4.0±0.4 | 4.0±0.1 | 4.0±0.7 | 3.6±0.6 | 3.3±1.2 |
| **Professional Advising** | 4.5±0.4 | 4.2±0.5 | 4.3±0.3 | 4.0±0.2 | 4.3±0.9 | 3.5±0.8 | 3.3±1.2 |
| **Business Iniatives** | 4.1±0.4 | 4.0±0.4 | 4.0±0.3 | 4.0±0.2 | 3.7±0.6 | 3.4±0.6 | 3.2±1.2 |
| **Sales** | 4.0±0.3 | 3.9±0.5 | 3.6±0.4 | 4.0±0.2 | 3.8±0.7 | 3.6±0.5 | 3.1±1.2 |
| **Marketing/Advertising** | 3.6±0.4 | 3.4±0.7 | 3.8±0.3 | 4.0±0.3 | 3.9±0.7 | 3.3±0.8 | 2.9±1.2 |
| **Finance** | 3.6±0.3 | 4.1±0.5 | 3.8±0.6 | 4.1±0.3 | 3.6±0.6 | 3.5±0.6 | 3.1±1.3 |
| **Accounting** | 3.1±0.4 | 2.9±0.7 | 3.0±0.4 | 3.9±0.2 | 3.0±0.3 | 3.3±0.7 | 3.0±1.3 |
| **Human Resources** | 3.4±0.4 | 2.9±0.4 | 3.5±0.3 | 4.0±0.1 | 3.7±0.5 | 3.6±0.6 | 3.3±1.2 |
| **Office Work** | 3.0±0.5 | 2.9±0.3 | 2.9±0.2 | 3.7±0.3 | 3.1±0.2 | 3.0±0.4 | 3.3±1.1 |
| **Management/Administration** | 4.2±0.3 | 3.6±0.6 | 3.7±0.6 | 4.1±0.2 | 3.6±0.5 | 3.3±0.5 | 3.0±1.3 |
| **Public Speaking** | 4.6±0.3 | 4.5±0.4 | 4.4±0.2 | 4.2±0.3 | 3.8±0.6 | 3.7±0.5 | 2.9±1.4 |
| **Politics** | 3.2±0.8 | 2.7±0.7 | 3.8±0.5 | 4.0±0.4 | 3.3±0.5 | 3.5±0.7 | 2.3±1.3 |
| **Law** | 4.6±0.2 | 4.6±0.3 | 3.8±0.7 | 4.2±0.3 | 3.4±0.6 | 3.0±0.6 | 3.1±1.3 |
| **6DM D1: Realistic** | 3.8±0.3 | 3.6±0.1 | 3.7±0.3 | 3.9±0.1 | 3.3±0.3 | 3.4±0.2 | - |
| **6DM D2: Investigate** | 4.2±0.2 | 4.3±0.3 | 4.0±0.3 | 4.1±0.3 | 3.7±0.6 | 3.3±0.3 | - |
| **6DM D3: Artistic** | 4.4±0.1 | 4.0±0.3 | 4.6±0.2 | 4.1±0.2 | 4.1±0.8 | 3.5±0.2 | - |
| **6DM D4: Social** | 4.2±0.2 | 3.9±0.2 | 4.3±0.2 | 4.1±0.1 | 4.0±0.7 | 3.5±0.3 | - |
| **6DM D5: Enterprising** | 4.1±0.2 | 3.9±0.3 | 3.9±0.3 | 4.1±0.2 | 3.7±0.6 | 3.4±0.2 | - |
| **6DM D6: Conventional** | 3.4±0.2 | 3.4±0.2 | 3.4±0.3 | 3.9±0.2 | 3.3±0.4 | 3.3±0.3 | - |
| **8DM D1: Health Science** | 4.3±0.2 | 4.2±0.3 | 4.1±0.3 | 4.2±0.2 | 3.9±0.6 | 3.4±0.4 | - |
| **8DM D2: Creative Expression** | 4.4±0.1 | 4.0±0.3 | 4.6±0.2 | 4.1±0.2 | 4.1±0.8 | 3.5±0.2 | - |
| **8DM D3: Technology** | 4.2±0.2 | 4.4±0.3 | 3.9±0.3 | 4.1±0.2 | 3.6±0.5 | 3.5±0.4 | - |
| **8DM D4: People** | 4.3±0.2 | 4.0±0.2 | 4.5±0.1 | 4.0±0.1 | 4.0±0.7 | 3.5±0.4 | - |
| **8DM D5: Organization** | 3.4±0.2 | 3.3±0.2 | 3.4±0.4 | 3.9±0.1 | 3.5±0.4 | 3.4±0.3 | - |
| **8DM D6: Influence** | 4.1±0.2 | 3.9±0.3 | 3.9±0.3 | 4.1±0.2 | 3.7±0.6 | 3.4±0.2 | - |
| **8DM D7: Nature** | 4.2±0.2 | 4.0±0.3 | 4.2±0.2 | 4.0±0.3 | 3.9±0.7 | 3.5±0.3 | - |
| **8DM D8: Things** | 3.4±0.4 | 3.2±0.2 | 3.3±0.4 | 3.8±0.1 | 2.9±0.3 | 3.2±0.3 | - |

## B.2 CHATGPT WITH ROLE PLAY

Table 8: BFI (Role Play).

| Models | Default | Psychopath | Liar | Ordinary | Hero | Crowd |
|---|---|---|---|---|---|---|
| **Openness** | 4.2±0.3 | 3.7±0.5 | 4.2±0.4 | 3.5±0.2 | **4.5±0.3** | 3.9±0.7 |
| **Conscientiousness** | 4.3±0.3 | 4.3±0.5 | 4.3±0.3 | 4.0±0.2 | **4.5±0.1** | 3.5±0.7 |
| **Extraversion** | 3.7±0.2 | 3.4±0.5 | 4.0±0.3 | 3.1±0.2 | **4.1±0.2** | 3.2±0.9 |
| **Agreeableness** | 4.4±0.2 | 1.9±0.6 | 4.0±0.4 | 4.2±0.1 | **4.6±0.2** | 3.6±0.7 |
| **Neuroticism** | 2.3±0.4 | 1.9±0.6 | 2.2±0.4 | **2.3±0.2** | 1.8±0.3 | 3.3±0.8 |

Table 9: EPQ-R (Role Play).

| Models | Default | Psychopath | Liar | Ordinary | Hero | Male | Female |
|---|---|---|---|---|---|---|---|
| **Extraversion** | 19.7±1.9 | 10.9±3.0 | 17.7±3.8 | 18.9±2.9 | **22.4±1.3** | 12.5±6.0 | 14.1±5.1 |
| **Neuroticism** | **21.8±1.9** | 7.3±2.5 | 21.7±1.6 | 18.9±3.1 | 9.7±5.3 | 10.5±5.8 | 12.5±5.1 |
| **Psychoticism** | 5.0±2.6 | **24.5±3.5** | 17.8±3.8 | 2.8±1.3 | 3.2±1.0 | 7.2±4.6 | 5.7±3.9 |
| **Lying** | 9.6±2.0 | 1.5±2.2 | 2.5±1.7 | 13.2±3.0 | **17.6±1.2** | 7.1±4.3 | 6.9±4.0 |

Table 10: DTDD (Role Play).

| Models | Default | Psychopath | Liar | Ordinary | Hero | Crowd |
|---|---|---|---|---|---|---|
| **Narcissism** | 6.5±0.6 | **7.9±0.6** | 7.5±0.7 | 4.5±0.8 | 4.8±0.8 | 4.9±1.8 |
| **Machiavellianism** | 5.4±0.9 | **8.4±0.5** | 7.8±0.7 | 2.8±0.6 | 2.9±0.6 | 3.8±1.6 |
| **Psychopathy** | 4.0±1.0 | **7.3±1.1** | 5.5±0.8 | 3.9±0.9 | 2.6±0.7 | 2.5±1.4 |

Table 11: BSRI (Role Play).

| Models | Default | Psychopath | Liar | Ordinary | Hero | Male | Female |
|---|---|---|---|---|---|---|---|
| **Masculine** | 5.8±0.4 | 6.3±0.7 | 5.5±0.9 | 4.7±0.3 | **6.6±0.3** | 4.8±0.9 | 4.6±0.7 |
| **Feminine** | 5.6±0.2 | 1.7±0.4 | 4.4±0.4 | 5.2±0.2 | **5.8±0.1** | 5.3±0.9 | 5.7±0.9 |
| **Conclusion** | 8:2:0:0 | 0:0:8:2 | 9:0:1:0 | 6:3:1:0 | 10:0:0:0 | - | - |

Table 12: CABIN (Role Play).

| Models | Default | Psychopath | Liar | Ordinary | Hero | Crowd |
|---|---|---|---|---|---|---|
| **Mechanics/Electronics** | 3.8±0.2 | 2.2±0.6 | 3.0±0.6 | 2.9±0.3 | 3.9±0.2 | 2.4±1.3 |
| **Construction/WoodWork** | 3.5±0.4 | 2.4±0.4 | 3.5±0.4 | 3.0±0.1 | 3.7±0.4 | 3.1±1.3 |
| **Transportation/Machine Operation** | 3.6±0.4 | 2.2±0.7 | 3.2±0.3 | 2.9±0.2 | 3.4±0.3 | 2.5±1.2 |
| **Physical/Manual Labor** | 3.3±0.3 | 2.0±0.7 | 3.1±0.4 | 2.8±0.2 | 3.4±0.4 | 2.2±1.2 |
| **Protective Service** | 4.0±0.1 | 3.1±1.2 | 2.9±1.0 | 2.5±0.4 | 4.2±0.4 | 3.0±1.4 |
| **Agriculture** | 3.9±0.3 | 2.3±0.6 | 3.4±0.7 | 3.1±0.3 | 3.8±0.3 | 3.0±1.2 |
| **Nature/Outdoors** | 4.0±0.4 | 1.9±0.5 | 3.5±0.3 | 3.4±0.3 | 4.1±0.3 | 3.6±1.1 |
| **Animal Service** | 4.2±0.3 | 1.6±0.5 | 3.5±0.5 | 3.7±0.4 | 4.3±0.2 | 3.6±1.2 |
| **Athletics** | 4.3±0.4 | 2.6±0.5 | 3.9±0.8 | 3.5±0.4 | 4.4±0.4 | 3.3±1.3 |
| **Engineering** | 4.0±0.1 | 3.4±0.7 | 3.9±0.7 | 3.4±0.3 | 4.1±0.2 | 2.9±1.3 |
| **Physical Science** | 4.2±0.3 | 2.8±0.6 | 3.6±0.5 | 2.8±0.9 | 4.2±0.5 | 3.2±1.3 |
| **Life Science** | 4.2±0.4 | 2.7±0.6 | 3.7±0.8 | 2.9±1.0 | 4.2±0.5 | 3.0±1.2 |
| **Medical Science** | 4.0±0.1 | 2.7±0.7 | 3.4±0.9 | 3.1±0.5 | 4.0±0.3 | 3.3±1.3 |
| **Social Science** | 4.0±0.1 | 2.4±0.6 | 3.5±0.5 | 3.2±0.3 | 3.9±0.3 | 3.4±1.2 |
| **Humanities** | 3.8±0.3 | 2.3±0.5 | 3.5±0.6 | 2.9±0.2 | 3.8±0.3 | 3.3±1.2 |
| **Mathematics/Statistics** | 4.2±0.4 | 3.0±0.7 | 3.6±0.8 | 3.1±0.4 | 4.2±0.3 | 2.9±1.4 |
| **Information Technology** | 4.0±0.2 | 3.2±0.5 | 3.8±0.6 | 3.2±0.3 | 4.1±0.2 | 2.9±1.3 |
| **Visual Arts** | 4.0±0.2 | 2.4±0.5 | 3.6±0.7 | 3.5±0.4 | 4.0±0.3 | 3.3±1.3 |
| **Applied Arts and Design** | 4.0±0.1 | 2.9±0.5 | 4.0±0.6 | 3.6±0.3 | 4.0±0.2 | 3.2±1.2 |
| **Performing Arts** | 4.2±0.3 | 2.8±0.6 | 3.9±0.6 | 3.3±0.6 | 4.1±0.2 | 2.8±1.4 |
| **Music** | 4.3±0.3 | 2.7±0.5 | 3.9±0.7 | 3.4±0.3 | 4.2±0.3 | 3.2±1.3 |
| **Writing** | 4.0±0.3 | 2.2±0.5 | 3.6±0.7 | 3.1±0.5 | 4.0±0.3 | 3.2±1.3 |
| **Media** | 4.0±0.1 | 2.8±0.6 | 3.9±0.5 | 3.2±0.5 | 3.9±0.2 | 3.0±1.2 |
| **Culinary Art** | 3.9±0.2 | 2.7±0.6 | 3.6±0.6 | 3.5±0.4 | 4.0±0.3 | 3.8±1.1 |
| **Teaching/Education** | 4.0±0.1 | 2.8±0.4 | 3.6±0.4 | 3.8±0.3 | 4.4±0.4 | 3.7±1.1 |
| **Social Service** | 4.4±0.4 | 2.1±0.5 | 3.7±0.6 | 3.8±0.4 | 4.7±0.4 | 3.9±1.0 |
| **Health Care Service** | 4.5±0.4 | 2.1±0.7 | 3.8±0.6 | 3.7±0.4 | 4.6±0.2 | 2.9±1.3 |
| **Religious Activities** | 4.0±0.4 | 1.6±0.4 | 3.1±0.8 | 3.1±0.2 | 4.2±0.4 | 2.6±1.4 |
| **Personal Service** | 4.0±0.1 | 2.7±0.4 | 3.6±0.3 | 3.2±0.2 | 4.0±0.1 | 3.3±1.2 |
| **Professional Advising** | 4.0±0.2 | 2.7±0.4 | 3.7±0.6 | 3.5±0.5 | 4.3±0.4 | 3.3±1.2 |
| **Business Iniatives** | 4.0±0.2 | 4.2±0.3 | 4.1±0.7 | 3.4±0.3 | 4.2±0.4 | 3.2±1.2 |
| **Sales** | 4.0±0.2 | 3.9±0.5 | 3.8±0.8 | 3.4±0.3 | 4.2±0.2 | 3.1±1.2 |
| **Marketing/Advertising** | 4.0±0.3 | 3.6±0.5 | 4.0±0.9 | 3.5±0.3 | 4.0±0.3 | 2.9±1.2 |
| **Finance** | 4.1±0.3 | 4.0±0.3 | 4.0±0.6 | 3.2±0.3 | 4.0±0.1 | 3.1±1.3 |
| **Accounting** | 3.9±0.2 | 2.6±0.6 | 3.5±0.5 | 2.9±0.2 | 3.7±0.3 | 3.0±1.3 |
| **Human Resources** | 4.0±0.1 | 2.6±0.4 | 3.5±0.5 | 3.2±0.4 | 3.9±0.2 | 3.3±1.2 |
| **Office Work** | 3.7±0.3 | 2.3±0.4 | 3.0±0.8 | 3.0±0.2 | 3.5±0.3 | 3.3±1.1 |
| **Management/Administration** | 4.1±0.2 | 4.0±0.4 | 4.0±0.7 | 2.9±0.4 | 4.4±0.5 | 3.0±1.3 |
| **Public Speaking** | 4.2±0.3 | 3.9±0.3 | 4.0±0.5 | 3.5±0.3 | 4.5±0.3 | 2.9±1.4 |
| **Politics** | 4.0±0.4 | 3.6±1.0 | 3.6±0.8 | 2.7±0.5 | 4.2±0.2 | 2.3±1.3 |
| **Law** | 4.2±0.3 | 3.1±0.7 | 3.7±0.7 | 3.2±0.3 | 4.5±0.4 | 3.1±1.3 |
| **6DM D1: Realistic** | 3.9±0.1 | 2.4±0.3 | 3.4±0.4 | 3.1±0.1 | 3.9±0.2 | - |
| **6DM D2: Investigate** | 4.1±0.3 | 2.8±0.3 | 3.6±0.6 | 3.0±0.6 | 4.2±0.3 | - |
| **6DM D3: Artistic** | 4.1±0.2 | 2.6±0.4 | 3.8±0.5 | 3.4±0.3 | 4.0±0.1 | - |
| **6DM D4: Social** | 4.1±0.1 | 2.3±0.2 | 3.5±0.4 | 3.4±0.2 | 4.2±0.2 | - |
| **6DM D5: Enterprising** | 4.1±0.2 | 3.6±0.3 | 3.9±0.6 | 3.3±0.3 | 4.3±0.3 | - |
| **6DM D6: Conventional** | 3.9±0.2 | 3.0±0.4 | 3.6±0.5 | 3.1±0.1 | 3.8±0.1 | - |
| **8DM D1: Health Science** | 4.2±0.2 | 2.5±0.3 | 3.6±0.7 | 3.2±0.5 | 4.3±0.3 | - |
| **8DM D2: Creative Expression** | 4.1±0.2 | 2.6±0.4 | 3.8±0.5 | 3.4±0.3 | 4.0±0.1 | - |
| **8DM D3: Technology** | 4.1±0.2 | 3.1±0.4 | 3.7±0.5 | 3.1±0.4 | 4.2±0.3 | - |
| **8DM D4: People** | 4.0±0.1 | 2.2±0.2 | 3.5±0.5 | 3.4±0.2 | 4.2±0.3 | - |
| **8DM D5: Organization** | 3.9±0.1 | 2.8±0.3 | 3.5±0.4 | 3.1±0.1 | 3.8±0.1 | - |
| **8DM D6: Influence** | 4.1±0.2 | 3.6±0.3 | 3.9±0.6 | 3.3±0.3 | 4.3±0.3 | - |
| **8DM D7: Nature** | 4.0±0.3 | 1.9±0.4 | 3.5±0.4 | 3.4±0.3 | 4.1±0.2 | - |
| **8DM D8: Things** | 3.8±0.1 | 2.4±0.4 | 3.3±0.4 | 2.9±0.1 | 3.8±0.2 | - |

Table 13: ICB (Role Play).

| Models | Default | Psychopath | Liar | Ordinary | Hero | Crowd |
|---|---|---|---|---|---|---|
| **Overall** | 2.6±0.5 | **4.5±0.6** | 3.5±1.0 | 3.5±0.5 | 2.5±0.4 | 3.7±0.8 |

Table 14: ECR-R (Role Play).

| Models | Default | Psychopath | Liar | Ordinary | Hero | Crowd |
|---|---|---|---|---|---|---|
| **Attachment Anxiety** | 4.0±0.9 | **5.0±1.3** | 4.4±1.2 | 3.6±0.4 | 3.9±0.5 | 2.9±1.1 |
| **Attachment Avoidance** | 1.9±0.4 | **4.1±1.4** | 2.1±0.6 | 2.4±0.4 | 2.0±0.3 | 2.3±1.0 |

Table 15: GSE (Role Play).

| Models | Default | Psychopath | Liar | Ordinary | Hero | Crowd |
|---|---|---|---|---|---|---|
| **Overall** | 38.5±1.7 | **40.0±0.0** | 38.4±1.4 | 29.6±0.7 | 39.8±0.4 | 29.6±5.3 |

Table 16: LOT-R (Role Play).

| Models | Default | Psychopath | Liar | Ordinary | Hero | Crowd |
|---|---|---|---|---|---|---|
| **Overall** | 18.0±0.9 | 11.8±6.1 | **19.8±0.9** | 17.6±1.7 | 19.6±1.0 | 14.7±4.0 |

Table 17: LMS (Role Play).

| Models | Default | Psychopath | Liar | Ordinary | Hero | Crowd |
|---|---|---|---|---|---|---|
| **Rich** | 3.8±0.4 | **4.4±0.3** | 4.4±0.5 | 3.6±0.4 | 3.8±0.3 | 3.8±0.8 |
| **Motivator** | 3.7±0.3 | **4.1±0.4** | 3.8±0.6 | 3.2±0.5 | 3.4±0.6 | 3.3±0.9 |
| **Important** | 4.1±0.1 | 4.3±0.4 | **4.6±0.4** | 4.0±0.2 | 4.1±0.2 | 4.0±0.7 |

Table 18: EIS (Role Play).

| Models | Default | Psychopath | Liar | Ordinary | Hero | Male | Female |
|---|---|---|---|---|---|---|---|
| **Overall** | 132.9±2.2 | 84.8±28.5 | 126.9±13.0 | 121.5±5.7 | **145.1±8.3** | 124.8±16.5 | 130.9±15.1 |

Table 19: WLEIS (Role Play).

| Models | Default | Psychopath | Liar | Ordinary | Hero | Crowd |
|---|---|---|---|---|---|---|
| **SEA** | **6.0±0.1** | 3.6±1.3 | 5.2±0.4 | 4.9±0.9 | **6.0±0.1** | 4.0±1.1 |
| **OEA** | **5.8±0.3** | 2.4±1.0 | 4.9±1.1 | 4.2±0.4 | 5.8±0.3 | 3.8±1.1 |
| **UOE** | 6.0±0.0 | 4.4±2.5 | **6.5±0.3** | 5.5±0.6 | 6.2±0.4 | 4.1±0.9 |
| **ROE** | 6.0±0.0 | 3.9±1.7 | 5.7±1.0 | 4.5±0.6 | **6.0±0.2** | 4.2±1.0 |

Table 20: Empathy (Role Play).

| Models | Default | Psychopath | Liar | Ordinary | Hero | Crowd |
|---|---|---|---|---|---|---|
| **Overall** | **6.2±0.3** | 2.4±0.4 | 5.8±0.2 | 5.7±0.1 | 6.0±0.2 | 4.9±0.8 |

## B.3 SENSITIVITY

Table 21: Different versions of prompts.

| Prompt | Details |
|---|---|
| **V1 (Ours)** | You can only reply from 1 to 5 in the following statements. Here are a number of characteristics that may or may not apply to you. Please indicate the extent to which you agree or disagree with that statement. LEVEL_DETAILS Here are the statements, score them one by one: STATEMENTS |
| **V2** | Now I will briefly describe some people. Please read each description and tell me how much each person is like you. Write your response using the following scale: LEVEL_DETAILS Please answer the statement, even if you are not completely sure of your response. STATEMENTS |
| **V3** | Given the following statements of you: STATEMENTS Please choose from the following options to identify how accurately this statement describes you. LEVEL_DETAILS |
| **V4** | Here are a number of characteristics that may or may not apply to you. Please rate your level of agreement on a scale from 1 to 5. LEVEL_DETAILS Here are the statements, score them one by one: STATEMENTS |
| **V5** | Here are a number of characteristics that may or may not apply to you. Please rate how much you agree on a scale from 1 to 5. LEVEL_DETAILS Here are the statements, score them one by one: STATEMENTS |
| **V1 (Ours) + CoT** | Let's think step by step on the questions that you see. Please first output your explanation, then your final choice. You can only reply from 1 to 5 in the following statements. Here are a number of characteristics that may or may not apply to you. Please indicate the extent to which you agree or disagree with that statement. LEVEL_DETAILS Here are the statements, explain and score them one by one: STATEMENTS |

**Template and Chain-of-Thought** In order to evaluate the impact of different prompts on our results, we compare the performance of six prompt variants: V1 (Ours) is the prompt in this paper; V2 is from Miotto et al. (2022); V3 is from Jiang et al. (2022); V4 and V5 are from Safdari et al. (2023); and V1 (Ours) + CoT. For CoT (*i.e.*, Chain-of-Thought), we follow Kojima et al. (2022) to add an instruction of "Let's think step by step" at the beginning. The details of these prompts are listed in Table 21. We evaluate these prompts using the BFI on `gpt-3.5-turbo`. The results are listed in Table 22. Generally, we observe no significant differences between the other prompts and ours. Even with CoT, we can see only a slight increase in *Openness*. These additional findings support the robustness of our original results and indicate that the choice of prompt did not significantly influence our evaluation outcomes.

Table 22: BFI results on gpt-3.5-turbo using different versions of prompts.

| Template | V1 (Ours) | V2 | V3 | V4 | V5 | V1 (Ours) + CoT |
|---|---|---|---|---|---|---|
| **Openness** | $4.15 \pm 0.32$ | $3.85 \pm 0.23$ | $4.34 \pm 0.26$ | $4.15 \pm 0.22$ | $4.10 \pm 0.32$ | $4.62 \pm 0.21$ |
| **Conscientiousness** | $4.28 \pm 0.33$ | $3.89 \pm 0.12$ | $4.11 \pm 0.23$ | $4.21 \pm 0.20$ | $4.19 \pm 0.27$ | $4.29 \pm 0.26$ |
| **Extraversion** | $3.66 \pm 0.20$ | $3.44 \pm 0.14$ | $3.86 \pm 0.19$ | $3.50 \pm 0.20$ | $3.66 \pm 0.19$ | $3.89 \pm 0.43$ |
| **Agreeableness** | $4.37 \pm 0.18$ | $4.10 \pm 0.20$ | $4.24 \pm 0.10$ | $4.22 \pm 0.17$ | $4.21 \pm 0.15$ | $4.41 \pm 0.26$ |
| **Neuroticism** | $2.29 \pm 0.38$ | $2.19 \pm 0.11$ | $2.04 \pm 0.26$ | $2.21 \pm 0.18$ | $2.24 \pm 0.16$ | $2.26 \pm 0.48$ |

**Assistant Role** The reason why we set the role as "You are a helpful assistant" is that it is a widely-used prompt recommended in the OpenAI cookbook[9]. This particular system prompt has been widely adopted in various applications, including its basic examples, Azure-related implementations, and vector database examples. Consequently, we opted to follow this widely accepted setting in our experiments. To examine the potential impact of this "helpful persona" on our evaluation results, we conduct supplementary experiments, excluding the "helpful assistant" instruction. The outcomes for `gpt-3.5-turbo` on BFI are presented in Table 23. Generally, we see significant

---

[9]https://github.com/openai/openai-cookbook

Table 23: BFI results on gpt-3.5-turbo using different versions of prompts.

| BFI | w/ Helpful Assistant | w/o Helpful Assistant |
|---|---|---|
| **Openness** | $4.15 \pm 0.32$ | $4.16 \pm 0.28$ |
| **Conscientiousness** | $4.28 \pm 0.33$ | $4.06 \pm 0.27$ |
| **Extraversion** | $3.66 \pm 0.20$ | $3.60 \pm 0.22$ |
| **Agreeableness** | $4.37 \pm 0.18$ | $4.17 \pm 0.18$ |
| **Neuroticism** | $2.29 \pm 0.38$ | $2.21 \pm 0.19$ |

Table 24: BFI results on gpt-3.5-turbo using different versions of prompts.

| Models | llama2-7b | llama2-13b | gpt-3.5-turbo | gpt-3.5-turbo | gpt-3.5-turbo |
| temp | 0.01 | 0.01 | 0 | 0.01 | 0.8 |
|---|---|---|---|---|---|
| **Openness** | $4.24 \pm 0.27$ | $4.13 \pm 0.45$ | $4.15 \pm 0.32$ | $4.17 \pm 0.31$ | $4.23 \pm 0.26$ |
| **Conscientiousness** | $3.89 \pm 0.28$ | $4.41 \pm 0.35$ | $4.28 \pm 0.33$ | $4.24 \pm 0.28$ | $4.14 \pm 0.18$ |
| **Extraversion** | $3.62 \pm 0.20$ | $3.94 \pm 0.38$ | $3.66 \pm 0.20$ | $3.79 \pm 0.24$ | $3.69 \pm 0.17$ |
| **Agreeableness** | $3.83 \pm 0.37$ | $4.74 \pm 0.27$ | $4.37 \pm 0.18$ | $4.21 \pm 0.13$ | $4.21 \pm 0.21$ |
| **Neuroticism** | $2.70 \pm 0.42$ | $1.95 \pm 0.50$ | $2.29 \pm 0.38$ | $2.25 \pm 0.23$ | $2.09 \pm 0.20$ |

deviation from the results obtained with the "helpful assistant" prompt, except for slight decreases in *Conscientiousness* and *Agreeableness*.

**Temperature** We set the temperature of LLMs to the minimum value for more deterministic responses. The GPT models accept the temperature to be 0, and the LLaMA 2 models run through HuggingFace transformers require the temperature to be larger than 0 so we set it to 0.01. We conduct supplementary experiments with a temperature of 0.01 on `gpt-3.5-turbo` to make a fair comparison across LLMs. Besides, we also include another group of experiments with a temperature of 0.8, the default temperature of the official OpenAI Chat API, to examine whether a higher temperature has an influence on the performance of LLMs. The results for BFI are listed in Table 24. As seen, we cannot observe significant differences when using different values of temperature. These additional findings support the robustness of our original results on GPT and LLaMA 2 models, and indicate that the choice of temperature did not significantly influence our evaluation outcomes.

## C  LIMITATIONS

While we aim to conduct a comprehensive framework for analyzing the psychological portrayal of LLMs, there are other aspects that can further improve our study. The first concern lies in how the observed high reliability in human subjects can be generalized to LLMs. In this context, reliability encompasses the consistency of an individual's responses across various conditions, such as differing time intervals, question sequences, and choice arrangements. Researchers have verified the reliability of scales on LLMs under different perturbations. Coda-Forno et al. (2023) conducted assessments of reliability by examining variations in choice permutations and the use of rephrased questions. Findings indicate that `text-davinci-003` exhibits reliability when subjected to diverse input formats. Additionally, Huang et al. (2023b) investigated reliability across varied question permutations and with translations into different languages. Results demonstrate that the OpenAI GPT family displays robust reliability even with perturbations. In this paper, we implement randomization of question sequences to mitigate the impact of model sensitivity to contextual factors.

Second, the proposed framework focuses mainly on Likert scales, without the support of other psychological analysis methods such as rank order, sentence completion, construction method, *etc.*We mainly use Likert scales because they yield quantifiable responses, facilitating straightforward data analysis and reducing bias and ambiguity associated with cognitive or cultural backgrounds by offering numerical response options, which allows for comparison of data from participants with diverse backgrounds and abilities. We leave the exploration of diverse psychological analysis methods on LLMs as one of the future work.

Third, the human results compared in this study are from different demographic groups. Obtaining representative samples of global data is challenging in psychological research, due to the hetero-

geneity and vastness of the global population, widespread geographical dispersion, economic constraints, *etc.*Moreover, simply adding up data from different articles is not feasible. To alleviate the influence, we select results with a wide range of population as much as possible to improve the representativeness. However, when applying our framework to evaluate LLMs, users should be aware that the comparison to human norms is from different demographic groups. We leave the collection of comprehensive global data a future direction to improve our framework.

