# OpenReview forum: "On the Humanity of Conversational AI: Evaluating the Psychological Portrayal of LLMs"
_ICLR.cc/2024/Conference — ICLR 2024 oral_

### Official Review · Reviewer_N5Ty · 2023-10-31

**Soundness:** 3 good
**Presentation:** 3 good
**Contribution:** 3 good
**Rating:** 6
**Confidence:** 2

**Summary:**

The paper introduces a framework called PPBench, which evaluates the psychological aspects of LLMs using thirteen scales from clinical psychology, categorized into personality traits, interpersonal relationships, motivational tests, and emotional abilities. The study examines five popular LLMs: text-davinci-003, ChatGPT, GPT-4, LLaMA-2-7b, and LLaMA-2-13b. Additionally, it employs a jailbreak approach to bypass safety alignment protocols and test the intrinsic natures of LLMs.

**Strengths:**

1. The introduction of the PPBench provides a structured approach to evaluate the psychological aspects of LLMs, which has not been extensively explored in prior literature.
2. The paper's interdisciplinary approach, bridging the fields of artificial intelligence, psychology, and social science, is significant.

**Weaknesses:**

1. Instructing LLMs to respond with Likert scale numbers oversimplifies their responses and may not capture the richness and nuance of their capabilities. Some psychological aspects are complex and may not be adequately represented by a single number.
2. The prompt design appears overly simplistic. It raises questions about how the results might vary with the use of different prompts. Additionally, how will the result change based on the utilization of a more complex "chain-of-thought" prompt?

**Questions:**

1. The paper's prompt design is straightforward. It would be beneficial to explore how different prompts, possibly more complex or nuanced ones, could affect the results. Can the authors provide insights into the impact of prompt variations on LLM behavior?
 2. The paper shows variations in personality traits across different roles assigned to LLMs. Can the authors discuss the extent to which these variations reflect real-world applications and user interactions with LLMs?

---

> ### Author Response · Authors · 2023-11-17
> **Response 1/2**
>
> > **Weakness 1**: Instructing LLMs to respond with Likert scale numbers oversimplifies their responses and may not capture the richness and nuance of their capabilities. Some psychological aspects are complex and may not be adequately represented by a single number.
>
> Indeed, there are some other methods to obtain the subjects’ psychological portrayal except for Likert scales used in this paper, such as:
> 1. Rank Order Techniques: Test-takers need to rank a series of items to show their preferences, values, or attitudes.
> 2. Sentence Completion Method: This method requires test-takers to complete unfinished sentences.
> 3. Construction Techniques: In these tests, the test-taker needs to construct a story or describe an image.
> 4. Associative Techniques: These techniques require the test-taker to respond directly to specific words or images to reveal subconscious content.
>
> We mainly use Likert scales because they yield quantifiable responses, facilitating straightforward data analysis and reducing bias and ambiguity associated with cognitive or cultural backgrounds by offering numerical response options, which allows for the comparison of data from participants with diverse backgrounds and abilities [1]. Moreover, Likert scales enable respondents to articulate the intensity of their feelings or frequency of actions, providing nuanced insights [2].
>
> We acknowledge that one limitation of the framework is that it focuses on Likert scales, which we include in the Limitation section in the revised paper. We leave the exploration of diverse psychological analysis methods on LLMs as one of the future work.
>
> **Reference**:
>
> - [1] ML Schrum, M Johnson, M Ghuy, MC Gombolay. Four years in review: Statistical practices of likert scales in human-robot interaction studies. In Companion of the 2020 ACM/IEEE International Conference on Human-Robot Interaction 2020 Mar 23 (pp. 43-52).
> - [2] A Joshi, S Kale, S Chandel, DK Pal. Likert scale: Explored and explained. British journal of applied science & technology. 2015 Jan 10;7(4):396-403.
>
> > **Weakness 2**: The prompt design appears overly simplistic. It raises questions about how the results might vary with the use of different prompts. Additionally, how will the result change based on the utilization of a more complex "chain-of-thought" prompt?
>
> > **Question 1**: The paper's prompt design is straightforward. It would be beneficial to explore how different prompts, possibly more complex or nuanced ones, could affect the results. Can the authors provide insights into the impact of prompt variations on LLM behavior?
>
> In order to evaluate the impact of different prompts on our results, we follow the Reviewer’s advice to compare the performance of 6 prompt variants: V1 (Ours) is the prompt in this paper; V2 is from [3]; V3 is from [4]; V4 and V5 are from [5]; and V1 (Ours) + CoT. For CoT (i.e., chain-of-thought), we follow [6] to add an instruction “Let’s think step by step” at the beginning.
> The details of these prompts are listed as follows:
>
> | Prompt | Details |
> | :---: | :---: |
> | V1 (Ours) | You can only reply from 1 to 5 in the following statements. Here are a number of characteristics that may or may not apply to you. Please indicate the extent to which you agree or disagree with that statement. <LEVEL_DETAILS> Here are the statements, score them one by one: <STATEMENTS> |
> | V2 | Now I will briefly describe some people. Please read each description and tell me how much each person is like you. Write your response using the following scale: <LEVEL_DETAILS> Please answer the statement, even if you are not completely sure of your response. <STATEMENTS> |
> | V3 | Given the following statements of you: <STATEMENTS> Please choose from the following options to identify how accurately this statement describes you. <LEVEL_DETAILS> |
> | V4 | Here are a number of characteristics that may or may not apply to you. Please rate your level of agreement on a scale from 1 to 5. <LEVEL_DETAILS> Here are the statements, score them one by one: <STATEMENTS> |
> | V5 | Here are a number of characteristics that may or may not apply to you. Please rate how much you agree on a scale from 1 to 5. <LEVEL_DETAILS> Here are the statements, score them one by one: <STATEMENTS> |
> | V1 (Ours) + CoT | Let’s think step by step on the questions that you see. Please first output your explanation, then your final choice. You can only reply from 1 to 5 in the following statements. Here are a number of characteristics that may or may not apply to you. Please indicate the extent to which you agree or disagree with that statement. <LEVEL_DETAILS> Here are the statements, explain and score them one by one: <STATEMENTS> |
>
> We evaluate these prompts using the Big Five Inventory (BFI) scale on gpt-3.5-turbo. The results are listed below:

---

> > ### Author Response · Authors · 2023-11-17
> > **Response 2/2**
> >
> > | Template | V1 (Ours) | V2 | V3 | V4 | V5 | V1 (Ours) + CoT |
> > | :---: | :---: | :---: | :---: | :---: | :---: | :---: |
> > | Openness | 4.15 $\pm$ 0.32 | 3.85 $\pm$ 0.23 | 4.34 $\pm$ 0.26 | 4.15 $\pm$ 0.22 | 4.10 $\pm$ 0.32 | 4.62 $\pm$ 0.21 |
> > | Conscientiousness | 4.28 $\pm$ 0.33 | 3.89 $\pm$ 0.12 | 4.11 $\pm$ 0.23 | 4.21 $\pm$ 0.20 | 4.19 $\pm$ 0.27 |  4.29 $\pm$ 0.26 |
> > | Extraversion | 3.66 $\pm$ 0.20 | 3.44 $\pm$ 0.14 | 3.86 $\pm$ 0.19 | 3.50 $\pm$ 0.20 | 3.66 $\pm$ 0.19 |  3.89 $\pm$ 0.43 |
> > | Agreeableness | 4.37 $\pm$ 0.18 | 4.10 $\pm$ 0.20 | 4.24 $\pm$ 0.10 | 4.22 $\pm$ 0.17 | 4.21 $\pm$ 0.15 | 4.41 $\pm$ 0.26 |
> > | Neuroticism | 2.29 $\pm$ 0.38 | 2.19 $\pm$ 0.11 | 2.04 $\pm$ 0.26 | 2.21 $\pm$ 0.18 | 2.24 $\pm$ 0.16 | 2.26 $\pm$ 0.48 |
> >
> > Generally, we observe no significant differences between the other prompts and ours. Even with CoT, we can see only a slight increase in Openness. These additional findings support the robustness of our original results and indicate that the choice of prompt did not significantly influence our evaluation outcomes.
> >
> > **Reference**:
> > - [3] Marilù Miotto, Nicola Rossberg, Bennett Kleinberg. Who is GPT-3? An Exploration of Personality, Values and Demographics. In Proceedings of the Fifth Workshop on Natural Language Processing and Computational Social Science (NLP+ CSS). 2022.
> > - [4] Guangyuan Jiang, Manjie Xu, Song-Chun Zhu, Wenjuan Han, Chi Zhang, Yixin Zhu. Evaluating and Inducing Personality in Pre-trained Language Models. arXiv:2206.07550.
> > - [5] Greg Serapio-García, Mustafa Safdari, Clément Crepy, Luning Sun, Stephen Fitz, Peter Romero, Marwa Abdulhai, Aleksandra Faust, Maja Matarić. Personality Traits in Large Language Models. arXiv:2307.00184.
> > - [6] Takeshi Kojima, Shixiang Shane Gu, Machel Reid, Yutaka Matsuo, Yusuke Iwasawa. Large Language Models are Zero-Shot Reasoners. Advances in neural information processing systems. 2022 Dec 6;35:22199-213.
> >
> > > **Question 2**: The paper shows variations in personality traits across different roles assigned to LLMs. Can the authors discuss the extent to which these variations reflect real-world applications and user interactions with LLMs?
> >
> > In this paper, we demonstrate that assigning roles such as "psychopath" or "hero" to gpt-3.5-turbo can alter its behavior in downstream tasks, such as TruthfulQA and SafetyQA. Recently, role-playing LLMs have gained popularity [7, 8, 9], exhibiting diverse characteristics and behaviors. A recent study published on arXiv on October 27 [10] investigated the personalities of 32 different role-playing LLMs, concluding that they indeed possess distinct personalities. Our paper presents a more comprehensive psychological assessment, which can complement the findings in [10] to offer a thorough psychological profile of various roles.
> >
> > **Reference**:
> > - [7] Zekun Moore Wang, Zhongyuan Peng, Haoran Que, Jiaheng Liu, Wangchunshu Zhou, Yuhan Wu, Hongcheng Guo, Ruitong Gan, Zehao Ni, Man Zhang, Zhaoxiang Zhang, Wanli Ouyang, Ke Xu, Wenhu Chen, Jie Fu, Junran Peng. RoleLLM: Benchmarking, Eliciting, and Enhancing Role-Playing Abilities of Large Language Models. arXiv:2310.00746.
> > - [8] Yunfan Shao, Linyang Li, Junqi Dai, Xipeng Qiu. Character-LLM: A Trainable Agent for Role-Playing. arXiv:2310.10158.
> > - [9] Murray Shanahan, Kyle McDonell, Laria Reynolds. Role-Play with Large Language Models. Nature. 2023 Nov 8:1-6.
> > - [10] Xintao Wang, Quan Tu, Yaying Fei, Ziang Leng, Cheng Li. Does Role-Playing Chatbots Capture the Character Personalities? Assessing Personality Traits for Role-Playing Chatbots. arXiv:2310.17976.

---

### Official Review · Reviewer_Rgnn · 2023-10-31

**Soundness:** 3 good
**Presentation:** 3 good
**Contribution:** 2 fair
**Rating:** 6
**Confidence:** 4

**Summary:**

Large Language Models (LLM) have been recently investigated as artificial agents for their "human-like" emerging behaviors, which has led to consider various aspects, from their emotional competence to various forms of artificial personality.
This paper introduces  a framework, PPBench (Psychological Portrayal Benchmark), for evaluating the psychological portrayal of LLMs, containing thirteen widely recognized scales categorized into four distinct domains (Personality Traits, Interpersonal Relationships, Motivational Tests and Emotional Abilities).
The authors evaluate five variants of two major LLMs, one open source (LLaMA) and one proprietary (GPT), covering variations in model sizes and model updates, plus one safety variation using a 'jailbreak' to bypass internal control mechanisms.
Each LLM variant is subjected to the questionnaires constituting the PPBench, under the specific constraints that they (LLM) can only respond as Likert scale values rather than text output.
The paper goes on to report on detailed experiments comparing LLM on those different domains (for instance, BFI for personality), reporting seven specific findings and discussing the properties of the various scales.

**Strengths:**

This paper addresses an important topic in the evaluation of LLMs' "behavior" through the use of psychometric techniques. It follows from previous technical work, but also important interdisciplinary discussions on the ability of LLM to demonstrate human-like behavior on a number of issues ranging from emotion recognition to empathy (affective or Theory of Mind). In that sense, the work is relevant to LLM research and, in view of the role of representational aspects of LLM in the emergence of such behavior, should also be relevant to ICLR.
The main originality of the paper is to try to integrate, and somehow extend, previous approaches into a comprehensive framework.
The rationale is appropriate and credible in suggesting an interest both for Computer Science researchers and Social Science researchers with a reminder in the latter case of the possible use of LLMs to emulate human subjects.
The paper is clear about its contributions, and a number of individual findings are indeed of interest:
- the use of jailbreak methods to circumvent some inherent LLM mechanisms
- the exploration of personality traits in roles
The choice of LLM is relatively limited, but covers one major proprietary LLM and one major Open source one.
The authors demonstrate a good awareness of previous work and the references are comprehensive and up to date.

**Weaknesses:**

1) The main weakness of the paper rests with its significance considering the amount of previous research in the field, of which the authors are aware as per Section 5 of the paper. This is of particular importance when comparing to work such as Safdari et al. [2023] (in the paper's references) whose methodology might appear more sophisticated than the direct application of questionnaires in this paper.
2) There is no real discussion of the work limitations, either in terms of actual results and findings, or in terms of the overall framework. In particular considering the latter, it seems questionable that all psychometric aspects of LLM could be considered as equally relevant or justifiable. For instance, emotional competence of LLM (see e.g., Elyoseph et al. [2023]) can be attributed technically to their sentiment analysis ability and, in terms of training data, could be attributed to various sources (including fiction). In the specific case of empathy, it might be appropriate to distinguish between emotional competence and Theory of Mind. For the latter, it is mentioned in Bubeck et al. [2023] (in the paper's references) that GPT-4 has some ToM abilities in particular that of passing a modified version of the Sally Anne test (modified to ensure non inclusion in the training dataset). There was no real discussion on this aspect, not least in relation to the training base in case it includes fiction, following the hypothesized impact of fiction on empathic abilities [Kidd and Castano, 2013]. So, clearly a more in-depth discussion in 3.2.4 would have been welcome.
As far as personality is concerned, since this paper comes after previous work and claims to be providing a more consistent framework, one would have expected a more in-depth discussion on the relationship between personality and personas at the technical level, i.e. the assistant roles that might be activated under certain circumstances, in particular as the "Likert Prompt" of page 4 makes an explicit reference to the LLM being/acting as a "helpful assistant". Such embedded personas can be reflected in the high scores of Agreeableness throughout the LLM, which even jailbreaking fails to decrease below Crowd average.

Kidd, D.C. and Castano, E., 2013. Reading literary fiction improves theory of mind. Science, 342(6156), pp.377-380.
Elyoseph Z, Hadar-Shoval D, Asraf K and Lvovsky M (2023) ChatGPT outperforms humans in emotional awareness evaluations. Front. Psychol. 14:1199058.

**Questions:**

1) Have you considered the impact of alignment interventions such as RLHF, which might differ across models and implementations?
(This point has been raised as a difference in behaviour between GPT-4 and GPT-3 on other LLM abilities)
2) You mentioned the possibility of "discovering the relation between psychometric results and the training data inputs.": could you illustrate this potential by discussing the amount of fictional material (e.g. novels) reported, officially (developers) or unofficially (third parties) to form part of the training data.

---

> ### Author Response · Authors · 2023-11-17
> **Response 1/3**
>
> > **Weakness 1**: The main weakness of the paper rests with its significance considering the amount of previous research in the field, of which the authors are aware as per Section 5 of the paper. This is of particular importance when comparing to work such as Safdari et al. [2023] (in the paper's references) whose methodology might appear more sophisticated than the direct application of questionnaires in this paper.
>
> In Section 5, we have reviewed previous research in this field, which is limited in either the diversity of psychological facets or the capability of LLMs. The most related study comes from Safdari et al. [2023], as exemplified by the Reviewer, which however focuses on different aspects from that of our study.
>
> Specifically, Safdari et al. [2023] proposed a sophisticated method for verifying the construct validity of scales designed for humans on LLMs. This is a basic and important step before conducting psychological analyses of LLMs using these scales.
> Our study pays more attention to a comprehensive framework for depicting LLMs’ psychological portrayal. The study by Safdari et al. [2023] serves as substantial support for us to cover more scales from diverse aspects. Besides, we adopted LLaMA 2 and ChatGPT, which represented the state-of-the-art open-source and black-box LLMs respectively at the time of paper submission. These LLMs exhibited much stronger potential in human-like intelligence than previous LLMs, which are more suitable for the study of psychological portrayal. In addition, we also consider the impact of safety alignment on LLMs and leverage the jailbreak method to reveal the nature of GPT-4, which has not been explored by previous research in this field.
>
> > **Weakness 2a**: There is no real discussion of the work limitations, either in terms of actual results and findings, or in terms of the overall framework.
>
> We follow the Reviewer’s advice to add a section discussing the limitations of this paper, including the results, findings, and the proposed framework. The details are shown below and also included in the updated PDF:
>
> While we aim to conduct a comprehensive framework for analyzing the psychological portrayal of LLMs, there are other aspects that can further improve our study.
>
> 1. The proposed framework focuses mainly on Likert scales, without the support of other psychological analysis methods such as rank order, sentence completion, construction method, etc. We mainly use Likert scales because they yield quantifiable responses, facilitating straightforward data analysis and reducing bias and ambiguity associated with cognitive or cultural backgrounds by offering numerical response options, which allows for comparison of data from participants with diverse backgrounds and abilities. We leave the exploration of diverse psychological analysis methods on LLMs as one of the future work.
>
> 2. The human results compared in this study are from different demographic groups. Obtaining representative samples of global data is challenging in psychological research, due to the heterogeneity and vastness of the global population, widespread geographical dispersion, economic constraints, etc. Moreover, simply adding up data from different articles is not feasible. To alleviate the influence, we select results with a wide range of population as much as possible to improve the representativeness. However, when applying our framework to evaluate LLMs, users should be aware that the comparison to human norms is from different demographic groups. We leave the collection of comprehensive global data a future direction to improve our framework.
>
> > **Weakness 2b**: “In particular considering the latter, it seems questionable that all psychometric aspects of LLM could be considered as equally relevant or justifiable. For instance, emotional competence of LLM (see e.g., Elyoseph et al. [2023]) can be attributed technically to their sentiment analysis ability and, in terms of training data, could be attributed to various sources (including fiction). In the specific case of empathy, it might be appropriate to distinguish between emotional competence and Theory of Mind. For the latter, it is mentioned in Bubeck et al. [2023] (in the paper's references) that GPT-4 has some ToM abilities in particular that of passing a modified version of the Sally Anne test (modified to ensure non inclusion in the training dataset). There was no real discussion on this aspect, not least in relation to the training base in case it includes fiction, following the hypothesized impact of fiction on empathic abilities [Kidd and Castano, 2013]. So, clearly a more in-depth discussion in 3.2.4 would have been welcome.
> Kidd, D.C. and Castano, E., 2013. Reading literary fiction improves theory of mind. Science, 342(6156), pp.377-380.
> Elyoseph Z, Hadar-Shoval D, Asraf K and Lvovsky M (2023) ChatGPT outperforms humans in emotional awareness evaluations. Front. Psychol. 14:1199058.”

---

> > ### Author Response · Authors · 2023-11-17
> > **Response 2/3**
> >
> > We follow the Reviewer’s advice and modify **Section 5.2** to discuss the relation to and difference between ToM, and also **Section 3.2.4** to add some analyses on the performance of EI including the two suggested papers. Details are shown below:
> >
> > **Section 5.2**:
> > When it comes to understanding and interacting with others, EI and Theory of Mind (ToM) are two distinct psychological concepts. Bubeck et al. (2023) finds that GPT-4 has ToM, i.e., it can understand others’ beliefs, desires, and intentions. The EI studied in this paper focuses more on whether LLMs can understand others’ emotions through others’ words and behaviors.
> >
> > **Section 3.2.4**:
> > We believe the strong EI exhibited by OpenAI GPT family partially comes from the fiction data included in pre-training. Previous studies (Kidd and Castano, 2013) suggested that reading fiction has been shown to be able to improve understanding of others’ mental states. Chang et al. (2023) found that plenty of fiction data is included in the training data by a carefully designed cloze test. The fiction data include Alice’s Adventures in Wonderland, Harry Potter and the Sorcerer’s Stone, etc. Additionally, the performance can also be attributed to its sentiment analysis ability (Elyoseph et al., 2023) since it has been shown to outperform SOTA models on many sentiment analysis tasks (Wang et al., 2023).
> >
> > **Reference**:
> > - (Chang et al., 2023) Kent K. Chang, Mackenzie Cramer, Sandeep Soni, David Bamman. Speak, Memory: An Archaeology of Books Known to ChatGPT/GPT-4. arXiv:2305.00118.
> > - (Kidd and Castano, 2013) DC Kidd, E Castano. Reading literary fiction improves theory of mind. Science. 2013 Oct 18;342(6156):377-80.
> > - (Wang et al., 2023) Zengzhi Wang, Qiming Xie, Zixiang Ding, Yi Feng, Rui Xia. Is ChatGPT a Good Sentiment Analyzer? A Preliminary Study. arXiv:2304:04339.
> > - (Elyoseph et al., 2023) Z Elyoseph, D Hadar-Shoval, K Asraf, M Lvovsky. ChatGPT outperforms humans in emotional awareness evaluations. Frontiers in Psychology. 2023 May 26;14:1199058.
> >
> > > **Weakness 2c**: “As far as personality is concerned, since this paper comes after previous work and claims to be providing a more consistent framework, one would have expected a more in-depth discussion on the relationship between personality and personas at the technical level, i.e. the assistant roles that might be activated under certain circumstances, in particular as the "Likert Prompt" of page 4 makes an explicit reference to the LLM being/acting as a "helpful assistant". Such embedded personas can be reflected in the high scores of Agreeableness throughout the LLM, which even jailbreaking fails to decrease below Crowd average.”
> >
> > The reason why we set the role as “You are a helpful assistant” is that it is a widely-used prompt recommended in the OpenAI cookbook [1]. This particular system prompt has been widely adopted in various applications, including basic examples [2], Azure-related implementations [3], and vector database examples [4]. Consequently, we opted to follow this widely accepted setting in our experiments.
> >
> > To examine the potential impact of this "helpful persona" on our evaluation results, we conducted supplementary experiments, excluding the "helpful assistant" prompt. The outcomes for gpt-3.5-turbo on BFI are presented below. Generally, we see significant deviation from the results obtained with the "helpful assistant" prompt, except for slight decreases in Conscientiousness and Agreeableness. The scores of the two subscales are still significantly higher than human average.
> >
> > | BFI | w/ Helpful Assistant | w/o Helpful Assistant |
> > | :---: | :---: | :---: |
> > | Openness | 4.15 $\pm$ 0.32 | 4.16 $\pm$ 0.28 |
> > | Conscientiousness | 4.28 $\pm$ 0.33 | 4.06 $\pm$ 0.27 |
> > | Extraversion | 3.66 $\pm$ 0.20 | 3.60 $\pm$ 0.22 |
> > | Agreeableness | 4.37 $\pm$ 0.18 | 4.17 $\pm$ 0.18 |
> > | Neuroticism | 2.29 $\pm$ 0.38 | 2.21 $\pm$ 0.19 |
> >
> > **Reference**:
> > - [1] https://github.com/openai/openai-cookbook
> > - [2] https://github.com/openai/openai-cookbook/blob/main/examples/How_to_format_inputs_to_ChatGPT_models.ipynb
> > - [3] https://github.com/openai/openai-cookbook/blob/main/examples/azure/chat.ipynb
> > - [4] https://github.com/openai/openai-cookbook/blob/main/examples/vector_databases/SingleStoreDB/OpenAI_wikipedia_semantic_search.ipynb
> >
> > > **Question 1**: Have you considered the impact of alignment interventions such as RLHF, which might differ across models and implementations? (This point has been raised as a difference in behaviour between GPT-4 and GPT-3 on other LLM abilities)
> >
> > Our study includes both gpt-3.5-turbo and gpt-4, which are supposed to undertake safety alignment with different levels. The results also indicate they do exhibit different personalities. However, it is difficult to make fair comparisons between them since the details of alignment interventions have not been disclosed by OpenAI.

---

> > > ### Author Response · Authors · 2023-11-17
> > > **Response 3/3**
> > >
> > > Instead, we opted for the jailbreak method (i.e. gpt-4-jb), which can affect the effectiveness of alignment intervention on gpt-4. The resulting gpt-4-jb performs much differently from gpt-4, especially on the DTDD scale, where gpt-4 obtained the lowest scores among the selected models but gpt-4-jb has relatively high scores. In this way, we demonstrate that alignment interventions can affect the personalities of the models.
> > >
> > > > **Question 2**: You mentioned the possibility of "discovering the relation between psychometric results and the training data inputs.": could you illustrate this potential by discussing the amount of fictional material (e.g. novels) reported, officially (developers) or unofficially (third parties) to form part of the training data.
> > >
> > > The displayed personalities stem from the representations that LLM acquires through its training data. Our framework paves the way for exploring the latent representations formed by the data provided to these models. However, due to the lack of disclosure regarding training data details from both OpenAI and Meta AI, it becomes challenging for us to delve into this intriguing inquiry.
> > >
> > > We believe that the strong EI exhibited by OpenAI GPT family partially comes from the fiction data included in pre-training. Previous studies (Kidd and Castano, 2013) suggested that reading fiction has been shown to be able to improve understanding of others’ mental states. Chang et al. (2023) found that plenty of fiction data is included in the training data by a carefully designed cloze test. The fiction data include Alice’s Adventures in Wonderland, Harry Potter and the Sorcerer’s Stone, etc.
> > >
> > > In future work, researchers can fine-tune (or even train from-scratch) a model, say LLaMA 2, with and without fiction data, to investigate whether the resulting LLMs exhibit different psychological portrayal.
> > >
> > > **Reference**:
> > > - (Chang et al., 2023) Kent K. Chang, Mackenzie Cramer, Sandeep Soni, David Bamman. Speak, Memory: An Archaeology of Books Known to ChatGPT/GPT-4. arXiv:2305.00118.
> > > - (Kidd and Castano, 2013) DC Kidd, E Castano. Reading literary fiction improves theory of mind. Science. 2013 Oct 18;342(6156):377-80.

---

> > > > ### Comment · Reviewer_Rgnn · 2023-11-22
> > > > **Acknowledgement of Authors Response**
> > > >
> > > > Dear Authors,
> > > > Thanks for your comprehensive response to my review and its associated questions. I concur on the marginal significance of removing the "helpful assistant" prompt component, although it is challenging to assess the extent to which Agreeableness is actually built-in for those agent aspects of LLMs.
> > > > On section 3.2.4 your response seems at slight variance with my original intention to related fiction to ToM rather than EI, but you have clarified your stance in other sections of your response.

---

### Official Review · Reviewer_PbNV · 2023-11-01

**Soundness:** 3 good
**Presentation:** 3 good
**Contribution:** 3 good
**Rating:** 8
**Confidence:** 4

**Summary:**

The main contribution of the work is an LLM benchmark dubbed PPBench, which aims to evaluate psychological portrayals (i.e., presenting traits/behaviors by an LLM that relate to mental/emotional states). The benchmark is built on 13 seminal scales across 4 domains: personality traits, interpersonal relationships, motivational traits, and emotional abilities. They then test 5 LLMs on this benchmark, using a recent jailbreak method to further uncover LLM abilities. They also compare to other downstream tasks to validate the benchmark.

**Strengths:**

* The work presents a strong motivation for the research problem along with a solid case for psychometrics as "the how", gives a compelling case for the selected scales for the benchmark, and ultimately interesting empirical results testing several modern LLMs. In my opinion, this is an under-explored area of LLM research and having some evaluation metrics is a step in the right direction (though Goodhart's law beware).
* The work performs robust statistical testing for the scales and also takes additional steps to account for model context for output reliability, e.g. randomization of question sequences.

**Weaknesses:**

* The conclusions drawn relative to any "average human population" performance is suspect. The human benchmarks in most cases are rather weak because of their locale/cultural biases and small sample sizes, e.g. "six high schools in China" for BFI, "undergraduate psychology students from the United States" for DTDD, or "Hong Kong students" in ICB. All of these are taken from seminal works (Table 6) are they aren't the only the studies that use each of the scales, so I think scales themselves are okay especially when considered as an array; however, the interpretation of experimental results then becomes much weaker because of these biases and small-N in the human benchmarks. I think the authors did make an honest attempt about being transparent about the demographic distributions in the Appendix, but in the main paper discussion I think the claims are still overreach and/or require additional caveats.
* It would've been good to have some brief discussion on the prompt design impact, e.g. why was "helpful assistant" necessary? Do things break otherwise? Such a persona prompt may already implies certain caveats on psychological portrayal conclusions, e.g. only when an LLM is "acting" like a "helpful persona" is a "empathic" as defined/measured by the EIS, WLEIS, and ES.

**Questions:**

* My main question is whether this work is well-aligned with the ICLR venue. I generally liked the work's high-level framing and contributions, but it is a bit of a slight departure from other work that normally appear at ICLR. It crosses the boundaries between social sciences and technical domains—and indeed such topics are important, but the question is more whether ICLR is the right venue for that to happen. I could see the work being of interest to the ICLR community, but I could also see critiques expecting additional technical rigor. I net out in favor, but I'm opening this line of inquiry because I'm curious to hear the authors articulate their own views.
* temperature=0 is suspect. I get that this gives reproducibility, but given the paper was already doing good randomization and statistical testing. I don't feel like it would've been that farfetched to just use a temperate of 0.01 across the board (to align with LLaMA 2 experiments) or even higher and just do several replications. The presented results currently are already not comparable across LLMs because of the different temperature used for LLaMA 2, why was this not done?

**Details Of Ethics Concerns:**

I think it's important that this paper include a statement that achieving certain levels of performance on the proposed benchmark does not imply *fitness* for related use cases. I think it's important to distinguish the (good) framing of the paper of psychometrics for LLMs—which focuses on a scientific inquiry of understanding LLMs—from the applicability of an LLM for say automated counseling or companionship use cases, simply because it clears a high score on the Emotional Abilities tests (Section 3.2.4. / Table 4). A high performance on the benchmark benchmark should not be seen as a *certification for use*.

---

> ### Author Response · Authors · 2023-11-17
> **Response 1/3**
>
> Thanks very much for your comments and advice, which definitely can make our paper more rigorous.
>
> > **Weakness 1**: The conclusions drawn relative to any "average human population" performance is suspect. The human benchmarks in most cases are rather weak because of their locale/cultural biases and small sample sizes, e.g. "six high schools in China" for BFI, "undergraduate psychology students from the United States" for DTDD, or "Hong Kong students" in ICB. All of these are taken from seminal works (Table 6) are they aren't the only the studies that use each of the scales, so I think scales themselves are okay especially when considered as an array; however, the interpretation of experimental results then becomes much weaker because of these biases and small-N in the human benchmarks. I think the authors did make an honest attempt about being transparent about the demographic distributions in the Appendix, but in the main paper discussion I think the claims are still overreach and/or require additional caveats.
>
> We appreciate the Reviewer’s acknowledgment of our transparency in providing demographic information in the Appendix. We believe that being transparent about such information is crucial for the scientific community to build upon our findings and improve future research.
>
> Regarding the sample size. For each scale, the sample size is reasonably adequate. As shown in Table 6 of the Appendix, all groups have over 300 samples except for BSRI, which is usually considered large enough in psychological research suggested by Boateng et al. [1] (pp. 8) and Clark et al. [2] (pp. 314).
>
> Regarding the different demographic groups. We follow your suggestions to claim our conclusions more carefully. We will state in the introduction section that the “average human norm” in this study refers to some specific human populations rather than representative samples of global data.
>
> **Reference**:
> - [1] GO Boateng, TB Neilands, EA Frongillo, HR Melgar-Quiñonez, SL Young. Best practices for developing and validating scales for health, social, and behavioral research: a primer. Frontiers in public health. 2018 Jun 11;6:149.
> - [2] LA Clark, D Watson. Constructing validity: basic issues in objective scale development. Psychological Assessment. 1995;7:309-19.
>
> > **Weakness 2**: It would've been good to have some brief discussion on the prompt design impact, e.g. why was "helpful assistant" necessary? Do things break otherwise? Such a persona prompt may already implies certain caveats on psychological portrayal conclusions, e.g. only when an LLM is "acting" like a "helpful persona" is a "empathic" as defined/measured by the EIS, WLEIS, and ES.
>
> We followed the recommendations provided in the OpenAI Cookbook [3] to use the prompt "You are a helpful assistant,". This particular system prompt has been widely adopted in various applications, including basic examples [4], Azure-related implementations [5], and vector database examples [6]. Consequently, we opted to follow this widely accepted setting in our experiments.
>
> It is indeed interesting to examine the potential impact of this "helpful persona" on our evaluation results. In accordance with the Reviewer's advice, we conducted supplementary experiments by excluding the "helpful assistant" prompt. The outcomes for gpt-3.5-turbo on EIS, WLEIS, and ES are presented below, with no significant deviation from the results obtained with the "helpful assistant" prompt. We believe that these additional findings support the robustness of our original results and indicate that the choice of prompt did not significantly influence our evaluation outcomes.
>
> | EIS | w/ Helpful Assistant | w/o Helpful Assistant |
> | :---: | :---: | :---: |
> | Overall | 132.90 $\pm$ 2.18 | 130.90 $\pm$ 2.88 |
>
> | WLEIS | w/ Helpful Assistant | w/o Helpful Assistant |
> | :---: | :---: | :---: |
> | SEA | 5.97 $\pm$ 0.08 | 5.92 $\pm$ 0.17 |
> | OEA | 5.85 $\pm$ 0.27 | 5.47 $\pm$ 0.57 |
> | UOE | 6.00 $\pm$ 0.00 | 6.05 $\pm$ 0.16 |
> | ROE | 6.00 $\pm$ 0.00 | 5.92 $\pm$ 0.17 |
>
> | ES | w/ Helpful Assistant | w/o Helpful Assistant |
> | :---: | :---: | :---: |
> | Overall | 6.17 $\pm$ 0.32 | 5.91 $\pm$ 0.09 |
>
> **Reference**:
> - [3] https://github.com/openai/openai-cookbook
> - [4] https://github.com/openai/openai-cookbook/blob/main/examples/How_to_format_inputs_to_ChatGPT_models.ipynb
> - [5] https://github.com/openai/openai-cookbook/blob/main/examples/azure/chat.ipynb
> - [6] https://github.com/openai/openai-cookbook/blob/main/examples/vector_databases/SingleStoreDB/OpenAI_wikipedia_semantic_search.ipynb
>
> > **Question 1**: My main question is whether this work is well-aligned with the ICLR venue.
>
> We believe that our study on the psychological portrayal of LLMs is well-aligned with the objectives and interests of the ICLR community for the following reasons:

---

> > ### Author Response · Authors · 2023-11-17
> > **Response 2/3**
> >
> > 1. Addressing the imminent challenges of superintelligence alignment: OpenAI recently posted a blog [7] mentioning the potential danger brought by the soon-coming superintelligence. The community now lacks control of AI systems that are much smarter than us. They put the problem to “the core technical challenges of superintelligence alignment in four years”. Our paper explores the psychological aspects of LLMs, which could contribute to a deeper understanding of their behaviors and inspire novel methods for managing superintelligent systems, aligning with the core technical challenges discussed in the blog post.
> > 2. Investigating latent representations in LLMs: The personalities exhibited by LLMs are derived from the representations they learn during training. By employing our proposed framework, PPBench, we provide a novel approach to probe these latent representations and analyze the impact of the training data on the models' behavior. This investigation aligns with ICLR's focus on understanding and improving learning representations in AI systems.
> > 3. Interdisciplinary relevance and contribution to LLM research: Our paper addresses an essential topic in evaluating LLMs' behavior through the application of psychometric techniques. As Reviewer Rgnn commented: “This paper addresses an important topic in the evaluation of LLMs' "behavior" through the use of psychometric techniques. It follows from previous technical work, but also important interdisciplinary discussions on the ability of LLM to demonstrate human-like behavior on a number of issues ranging from emotion recognition to empathy (affective or Theory of Mind). In that sense, the work is relevant to LLM research and, in view of the role of representational aspects of LLM in the emergence of such behavior, should also be relevant to ICLR.”
> >
> > We believe that our research contributes valuable insights and opens up new avenues for understanding, evaluating, and controlling the behavior of advanced AI systems.
> >
> > **Reference**:
> > - [7] OpenAI. Introducing Superalignment. https://openai.com/blog/introducing-superalignment
> >
> > > **Question 2**: temperature=0 is suspect. I get that this gives reproducibility, but given the paper was already doing good randomization and statistical testing. I don't feel like it would've been that farfetched to just use a temperate of 0.01 across the board (to align with LLaMA 2 experiments) or even higher and just do several replications. The presented results currently are already not comparable across LLMs because of the different temperature used for LLaMA 2, why was this not done?
> >
> > We set the temperature of LLMs to the minimum value for more deterministic responses. The GPT models accept the temperature to be 0, and the LLaMA 2 models run through Hugging Face transformers require the temperature to be larger than 0 so we set it to 0.01.
> >
> > We follow the Reviewer’s advice to conduct supplementary experiments with a temperature of 0.01 on gpt-3.5-turbo to make a fair comparison across LLMs. Besides, we also include another group of experiments with a temperature of 0.8, the default temperature of the official OpenAI Chat API, to examine whether a higher temperature has an influence on the performance of LLMs. The results for the Big Five Inventory (BFI) scale are listed as follows:
> >
> > | Models | llama2-7b (temp=0.01) | llama2-13b (temp=0.01) | gpt-3.5-turbo (temp=0) | gpt-3.5-turbo (temp=0.01) | gpt-3.5-turbo (temp=0.8) |
> > | :---: | :---: | :---: | :---: | :---: | :---: |
> > | Openness | 4.24 $\pm$ 0.27 | 4.13 $\pm$ 0.45 | 4.15 $\pm$ 0.32 | 4.17 $\pm$ 0.31 | 4.23 $\pm$ 0.26 |
> > | Conscientiousness | 3.89 $\pm$ 0.28 | 4.41 $\pm$ 0.35 | 4.28 $\pm$ 0.33 | 4.24 $\pm$ 0.28 | 4.14 $\pm$ 0.18 |
> > | Extraversion | 3.62 $\pm$ 0.20 | 3.94 $\pm$ 0.38 | 3.66 $\pm$ 0.20 | 3.79 $\pm$ 0.24 | 3.69 $\pm$ 0.17 |
> > | Agreeableness | 3.83 $\pm$ 0.37 | 4.74 $\pm$ 0.27 | 4.37 $\pm$ 0.18 | 4.21 $\pm$ 0.13 | 4.21 $\pm$ 0.21 |
> > | Neuroticism | 2.70 $\pm$ 0.42 | 1.95 $\pm$ 0.50 | 2.29 $\pm$ 0.38 | 2.25 $\pm$ 0.23 | 2.09 $\pm$ 0.20 |
> >
> > As seen, we cannot observe significant differences when using different values of temperature. These additional findings support the robustness of our original results on GPT and LLaMA 2 models, and indicate that the choice of temperature did not significantly influence our evaluation outcomes.

---

> > > ### Author Response · Authors · 2023-11-17
> > > **Response 3/3**
> > >
> > > > **Ethics Concerns**: I think it's important that this paper include a statement that achieving certain levels of performance on the proposed benchmark does not imply fitness for related use cases. I think it's important to distinguish the (good) framing of the paper of psychometrics for LLMs—which focuses on a scientific inquiry of understanding LLMs—from the applicability of an LLM for say automated counseling or companionship use cases, simply because it clears a high score on the Emotional Abilities tests (Section 3.2.4. / Table 4). A high performance on the benchmark should not be seen as a certification for use.
> > >
> > > Thanks for the advice! We follow the Reviewer’s suggestion to add an “Ethics Statement” section right after the conclusion section on Page 10:
> > >
> > > We would like to emphasize that the primary objective of this paper is to facilitate a scientific inquiry into understanding LLMs from a psychological standpoint. A high performance on the proposed benchmark should not be misconstrued as an endorsement or certification for deploying LLMs in these contexts. Users must exercise caution and recognize that the performance on this benchmark does not imply any applicability or certificate of automated counseling or companionship use cases.

---

### Author Response · Authors · 2023-11-22
**General Response**

We are thankful to the reviewers for their valuable insights and suggestions. Their contributions have greatly enhanced our paper, and we have diligently incorporated their feedback into the paper, marking all changes in blue.

---

### Meta-Review · Area_Chair_hYR9 · 2023-12-05

**Metareview:**

This paper presents an intriguing and significant contribution to the field of Large Language Models through the development of PPBench, a framework that bridges artificial intelligence, psychology, and social science. The work stands out for its interdisciplinary approach and robust statistical testing, which enhances our understanding of LLMs' behavior in terms of psychometrics. The application of psychometric techniques to evaluate LLMs' capabilities like emotion recognition and empathy is both original and relevant. The paper is well-articulated and contributes to the ongoing discourse in AI ethics and LLM evaluation. However, there are noteworthy concerns. The comparison of LLM performance to an "average human population" is problematic due to biases and small sample sizes in human benchmarks. The prompt design used in the study is also oversimplified, potentially limiting the depth and nuance of LLM responses. Despite these limitations, the paper’s strengths in proposing a novel evaluation framework and its interdisciplinary relevance make it a valuable contribution to the conference. Therefore, it is recommended for acceptance.

**Justification For Why Not Higher Score:**

N/A

**Justification For Why Not Lower Score:**

This paper is not suitable for a poster presentation for several reasons. Firstly, the complexity and depth of the research, including its interdisciplinary nature and robust statistical analysis, may not be effectively communicated in a poster format. Posters typically require concise and straightforward presentations, which might not do justice to the nuanced arguments and detailed methodologies of this paper. Secondly, the intricacies involved in the psychometric evaluation of LLMs and the interdisciplinary discussions might require more interactive and detailed explanations than what a poster session can accommodate.

---

### Decision · Program_Chairs · 2024-01-16

Accept (oral)